# SimpleStrat: Diversifying Language Model Generation with Stratification

**Justin Wong**
UC Berkeley
wong.justin@berkeley.edu

**Yury Orlovskiy**
UC Berkeley
yury.orlovskiy@berkeley.edu

**Alexander Shypula**
University of Pennsylvania
shypula@seas.upenn.edu

**Michael Luo**
UC Berkeley
michael.luo@berkeley.edu

**Sanjit A. Seshia**
UC Berkeley
sseshia@berkeley.edu

**Joseph E. Gonzalez**
UC Berkeley
jegonzal@berkeley.edu

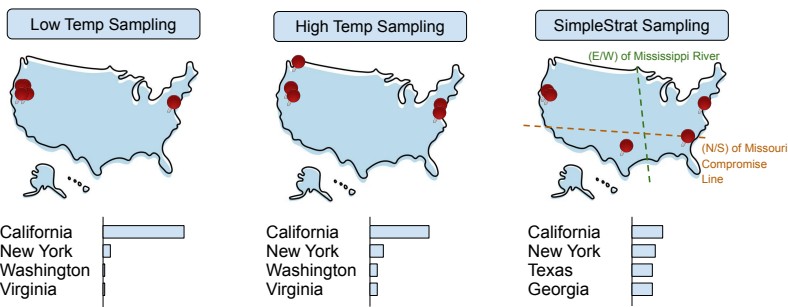

Figure 1: **SimpleStrat recovers diversity when temperature scaling fails.** For the prompt "Name a U.S. state," language models exhibit a strong bias toward "California." Our method, SimpleStrat, employs automatic stratification to uncover salient diversity dimensions (e.g., whether a state lies east or west of the Mississippi River) and applies stratified sampling to achieve balanced and diverse coverage across responses.

## Abstract

Generating diverse responses from large language models (LLMs) is crucial for applications such as adversarial testing, search, and synthetic data generation, where diversity provides distinct answers across generations. Previous approaches rely solely on increasing the temperature, sacrificing quality. Furthermore, the model's next-token probabilities may not be representative of the true answer distribution. To combat these challenges, we propose SimpleStrat, an alternative that uses the language model itself to partition the solution space into strata from which to sample. To measure resampling diversity, we introduce CoverageQA, a dataset of underspecified questions with multiple equally plausible answers. We propose measuring resampling diversity as the KL Divergence between the response distribution and the uniform distribution over valid ground truth answers and use recall as an alternative when assessing proprietary models. On CoverageQA, SimpleStrat improves diversity across all temperatures, showing orthogonal benefits. Quantifiably, we achieve as much as 4X better recall when applied to GPT-4o, and an average reduction in KL divergence by 0.36 when applied to Llama 3. Furthermore, we show that SimpleStrat achieves more resampling diversity at temperature T=0 than scaling temperature to T=1 on creative writing, an open-ended domain. Implementation and dataset available at `https://github.com/jwong8314/simplestrat` .

39th Conference on Neural Information Processing Systems (NeurIPS 2025).

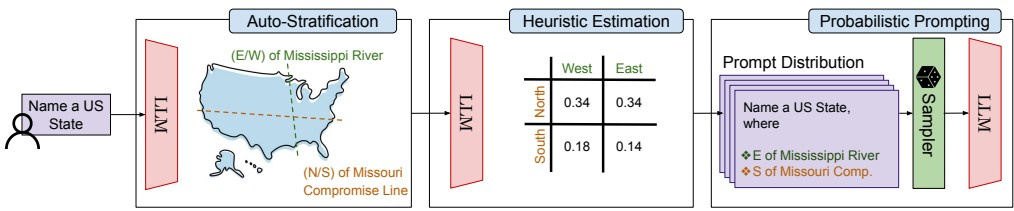

Figure 2: **SimpleStrat workflow.** SimpleStrat employs 3 phases: 1) auto-stratification to identify good dimensions of diversity that divide the solution space into equal partitions, 2) heuristic estimation to estimate the proportion of solutions in each stratum, and 3) probabilistic prompting where a concrete prompt is randomly sampled from the prompt distribution specified by the previous two phases. Critically, diverse resampling comes from both the random choice of prompt as well as the temperature of the LLM decoding.

# 1   Introduction.

Large language models (LLMs) are routinely resampled in order to get a wide set of plausible generations. Three key settings where this is important are: 1) improving exploration for planning and search (e.g. Tree-of-thought (Yao et al., 2024), AgentQ (Putta et al., 2024)), 2) generating inputs for integration tests, especially for LLM systems Samvelyan et al. (2024), and 3) generating diverse datasets for post-training (Dubey et al., 2024) and fine-tuning (Dai et al., 2023). These use cases rely on resampling outputs to generate multiple plausible utterances in hopes of capturing multiple answers and candidate solutions.

Temperature scaling, which controllably flattens an LLM's softmax distribution, is commonly used to increase response diversity. However, this approach assumes that next-token probabilities accurately represent the set of valid answers, which is not true under mode collapse. In such cases, increasing temperature does not lead to meaningful diversity. Moreover, temperature scaling introduces a second limitation: higher temperatures degrade generation quality. Recent evidence suggests that removing temperature scaling is desirable for multi-step reasoning to reduce error compounding (Zhang et al., 2024). This is especially critical in syntax-sensitive settings such as code generation, where low temperatures ($\leq 0.15$) are often used. We illustrate this severe mode collapse in Figure 1. When asked to "Name a US State," the model heavily skews towards answering "California", high temperature only marginally softens the skew while surfacing incorrect answers and hurting instruction following.

Our goal is to improve response diversity when resampling LLMs, even in cases of severe mode collapse Lanchantin et al. (2025) in next-token probabilities without manual intervention. Our analysis reveals that GPT-4 assigns 87% of its logit weight to "California" when prompted to name a US state. This observed bias can be attributed to the worsening of calibration due to post-training as reported in the GPT-4 tech report (OpenAI et al., 2024). This stark bias mirrors human cognitive bias, exemplified by the blue-seven phenomenon—where individuals disproportionately select blue and seven when asked to choose a random color and number (Towse et al., 2014). To counteract similar biases in human populations, social scientists, particularly in political polling, employ stratified sampling techniques (Simpson, 1951; Howell, 1992; Morris, 2022). We propose adapting this method to address mode collapse in LLMs.

We propose SimpleStrat, a training-free sampling approach to increase diversity of responses. SimpleStrat improves LLM generation diversity without compromising quality, yielding outputs that better align with the task's objective answer distribution. SimpleStrat consists of three stages: auto-stratification, heuristic estimation, and probabilistic prompting. Even if a language model cannot generate diverse solutions, we find that it can be prompted to identify useful partitions of the solution space based on the user request. We call this process *auto-stratification*. In Fig. 1, SimpleStrat identifies two semantically significant strata from user request, "Name a US State": "(East/West) of the Mississippi River" and "(North/South) of the Missouri Compromise Line."

Next, the heuristic estimation computes the joint probabilities across all strata. Returning to Fig. 1, SimpleStrat estimates the proportion of solutions in each four possible regions of the US. Finally, SimpleStrat samples a stratum from the joint probability distribution and augments the original user prompt with constraints based on the selected stratum. This prompt can then be used to sample fully-formed answers to the user query. We note that this approach to repsonse diversity is orthogonal to increasing temperature which also introduces variance to the generation.

Response diversity is notoriously difficult to measure as it requires a notion of quality specific to the application. Otherwise, random tokens present a trivial solution to achieving diverse yet meaningless responses. On the extreme of measuring quality, question answering benchmarks capture the quality of the responses but are carefully designed to only admit one valid solution. As such, a fitting setting to measure diversity is required to have underspecified questions for which more than one plausible answer exists. Distinct from ambiguous question answering where the goal is to measure the model's ability to ask clarifying questions, we seek questions where the quality of the response can be independently verified without additional context. Based on these requirements, we introduce CoverageQA, a benchmark of underspecified questions with on average 28.7 equally plausible answers.

We adopt three measures of response diversity based on the setting: 1) In settings where the target answer distribution and logits are available, we measure *distributional diversity* by computing the Kullback-Leibler (KL) Divergence from the response distribution to the target answer distribution over valid answers. By measuring diversity this way, we can both penalize models for producing nonsense responses and missing valid responses. 2) In settings where we have only black-box access to the model, the response distribution can be estimated by sampling. Since the tail of the distribution is inherently difficult to capture, recall and precision serve as a natural proxy for diversity and quality. We call this *coverage diversity*. 3) Finally, in settings where exhaustively enumerating valid solutions is infeasible, we cannot assess if valid responses are missing. As such, we propose text embedding distance as a proxy for diversity. We assess SimpleStrat across these three settings: measuring *distributional diversity* with Llama 3 and *coverage diversity* over Claude and GPT-4o with prompts from CoverageQA as well as *open-ended diversity* with embedding distance on creative writing prompts from WritingPrompts Fan et al. (2018).

When assessing *distributional diversity*, we show SimpleStrat samples from a less biased distribution by computing the response distribution of our method based on next-token probabilities. On CoverageQA, SimpleStrat implemented on Llama 3 models demonstrate a substantial 0.36 average reduction in KL Divergence, signifying a substantial decrease in dissimilarity between the model's response distribution and the target distribution. In the blackbox setting, we see as much as $4\times$ increase in recall, especially evident at low temperatures. These gains lead to a consistent increase in diversity on top of temperature scaling, leading to improved diversity at all temperatures. Finally, we consider the open-ended setting of creative writing. We measure diversity by resampling pairs of creative writing story plots from WritingPrompts Fan et al. (2018) and see similar embedding distances at temperature 0 sampling with SimpleStrat comparable to temperature 1 with standard decoding. Crucially, these gains do not come at the expense of generation quality.

Concretely, our work contributes the following:

- CoverageQA dataset of 155 underspecified questions automatically generated from Wiki-Data (Vrandečić & Krötzsch, 2014) automatically labeled with an average of 24.1 valid solutions per question.
- We propose SimpleStrat a training-free approach for improving diversity with *auto-stratification* and *probabilistic prompting* by introducing controlled randomness to the prompting process.
- We demonstrate SimpleStrat improves diversity across several metrics and settings. On CoverageQA, we achieve an average of 0.36 reduction in KL Divergence on Llama 3 models and as much as $4\times$ recall for GPT-4o. Further on WritingPrompts Fan et al. (2018), SimpleStrat shows similar pairwise embedding distance at temperature 0 as temperature 1 with standard sampling.

## 2 Related work.

**Temperature Scaling.** Going back as far as Platt scaling (Platt, 2000) and later applied to neural networks (Hinton, 2015; Guo et al., 2017), temperature scaling controls the randomness of probability distributions[1]. For dataset generation with LLMs, Chung et al. (2023) extends temperature-based diversity by additionally downsampling previously sampled tokens. To address the decrease in quality, they advocate for human intervention to manually filter out irrelevant diversity and manually fix wrong answers in QA tasks. We show in our work that temperature scaling leaves much to be desired.

---

[1]Use of a temperature parameter goes back at least to Verhulst's development of logistic regression in response to Malthus' *An Essay on Principle of Population* (Malthus, 1798; Verhulst, 1838).

**Improving Language Model Diversity with Search.** In autoregressive generation, choices over early tokens tend to have more impact on the eventual completion. Beam search ameliorates this bias by allowing for multiple candidates in searching for the probability maximizing completion, Maximum a Posteriori (Lowerre & Reddy, 1976). At the end of the search, beam search will have multiple candidate solutions encountered during search. Diverse Beam Search (DBS) proposes introducing an auxiliary dissimilarity objective quantifying the diversity among candidates in the beam (Vijayakumar et al., 2016). Especially on the task of image captioning, DBS shows improvement for discovering higher probability completions and discovering diverse continuations. Our improvements are orthogonal to beam search and our in-context approach corrects for inaccuracies in the modeled likelihoods of candidate solutions.

Other approaches (Samvelyan et al., 2024; Bradley et al., 2023; Novikov et al., 2025) based on MAP-Elites (Mouret & Clune, 2015) require manual determined dimensions of relevant diversity and discretization of the solution space into equally-sized bins. Diversity is then achieved by mutations and evolutionary methods to cover adjacent bins. This search is potentially slow if the seed set of solutions does not already provide coverage over the solutions space. Our approach does not need seed solutions and avoids manually identifying dimensions of diversity. Instead, we rely solely on capabilities within the model.

**In-context Methods to Increase Diversity.** When LLMs were first introduced, LMs were used to augment existing datasets with more diversity (Wei & Zou, 2019; Ng et al., 2020; Dai et al., 2023). As natural language is difficult to guarantee correctness, the space of augmentations is conservatively limited to thesaurus-based synonym replacement. More recently, Language Model Crossover proposes presenting a random subset of existing data points to an LLM and ask it to hallucinate more data points that likely came from the same distribution Meyerson et al. (2023). This is limited to combining aspects of existing data points into new generations. Although these methods address the limitations of using the model's token probabilities by in-context learning, they are ineffective at generating meaningful diversity. They are limited to either a human-identified domains of interest or trivial variations sourced from synonyms or random subsampling of the existing data.

**Applications of Diversity.** As shown by Raventós et al. (2024), dataset diversity is crucial for model generalization. Below sufficient coverage of the desired task, the model will resort to memorization, but when sufficient diversity is presented it will learn to generalize. As LLMs are increasingly used for generating synthetic data (Dubey et al., 2024), methods for diversity will be critical. This insight follows from extensive work demonstrating the benefits of data augmentation for bias mitigation (Sharmanska et al., 2020) and domain adaptation (Huang et al., 2018; Dunlap et al., 2023; Trabucco et al., 2023).

In code and math applications, checking validity efficiently enables more aggressive augmentations. One such augmentation for diversifying the languages supported by the model, data is translated to different natural or programming language (Chen et al., 2023; Cassano et al., 2023). In other domains such as images, text-to-image models have been used to diversify data into uncommon settings. In the setting of diversifying an accumulating dataset, these methods can take advantage of an existing source of variance (for translation) or a set of previously generated data points. Our primary focus is on settings where SimpleStrat is unaware of past data samples to support a wider set of applications.

**Ambiguous or Underspecified Datasets.** ClariQ (Aliannejadi et al., 2020), CLAQUA (Xu et al., 2019), and AmbigQA (Min et al., 2020) focus on assessing LM's ability to formulate clarifying questions. These questions tend to have only two candidate solutions, as there exists a ground truth clarifying question whose answer fully specifies the question. Ambiguous Trivia QA (Kuhn et al., 2022) also looks at under-specified questions but assumes a user has contextual information that's hidden. For instance, "Where in England was she born?" or "Who was the first woman to make a solo flight across this ocean?". We distinguish our underspecified question setting in this paper as one where the user is indifferent. In this setting, given an answer it should be easy to verify the answer is correct without additional hidden context.

Coding datasets like Description2Code (Caballero et al., 2016), Wiki2SQL (Zhong et al., 2017), SPIDER (Yu et al., 2019), code-contest (Li et al., 2022), Apps (Hendrycks et al., 2021), and Leetcode Hard Shinn et al. (2023) admit multiple valid answers. However, the space of valid implementations is infinite, making diversity difficult to measure, and good coding practices enforce preferences among valid implementations. We additionally construct CoverageQA to have an exhaustive list of ground-truth answers in order to measure the impact of diversity on coverage.

# 3 Method

## 3.1 Workflow overview

As illustrated in Figure 2, SimpleStrat consists of three stages, 1) auto-stratification, 2) heuristic estimation, and 3) probabilistic prompting. The outputs of the first two stages can be cached per prompt to avoid recomputing partitions.

## 3.2 Auto-Stratification

For a given user request, $r_{user}$, we call $S$, the space of valid solutions. In many settings, the space of potential solutions, $S$ may be naturally partitioned based on geography, parity, or demographics. The partition function, $P : S \rightarrow L$, assigns any solution $s$ from $S$ to a partition label $l_j$ in $L$ the set of partition labels. Partition functions are most useful if they're as balanced as possible. A balanced partition function minimizes

$$imbalance(P, L) = \max_{l \in L} (|\{s \mid P(s) = l\}|) - \min_{l \in L} (|\{s \mid P(s) = l\}|)$$

The goal of auto-stratification is to search for a set of partition functions $\mathbf{P} = \{P_1, P_2, ..., P_n\}$, that are balanced. In settings where valid solutions are easy to miss or the solution space is large (or even infinite), stratified sampling ensures that a limited number of samples provides broad and even coverage.

Based on this insight, we prompt the language model to identify promising dimensions of diversity. Concretely, the language model proposes good clarifying questions that will potentially eliminate half of the potential solutions based on the user request. These clarifying questions tend to align with semantically significant differences. In the running example, when asked, "Name a US State," the states can be partitioned based on East or West of the Mississippi River. See App. E for full prompt.

## 3.3 Heuristic estimation

As previously observed in Zou et al. (2022); Yan et al. (2023); Halawi et al. (2024), LLMs can used in forecasting to estimate well-calibrated probabilities of events that have not yet occured. For forecasting, the model's success benefits substantially from having updated news through web search. Although unnecessary for the offline benchmarks we consider, this may be helpful for accurate estimation depending on the application. However, as our goal is diversity, we stand to benefit even from coarse-grain approximate proportions. We employ a similar reasoning template as Halawi et al. (2024) to estimate the proportion of valid solutions that lies within each strata.

In heuristic estimation, we look to estimate the joint distribution for each stratum, $\vec{l} = [l_1, l_2, l_3, ...]$. Formally, we define the weighted-stratification as $\mathcal{W} = (\mathbf{P}, \rho)$, where $\rho(\vec{l}) = Pr_{s \sim S}[P_1(s) = l_{1,j}, P_2(s) = l_{2,j}, P_3(s) = l_{1,j}, ...]$ for $\mathbf{P}$ identified in auto-stratification. To improve scalability, we assume the partition functions are independent and multiply the marginal probabilities to get the joint probabilities associated with each stratum.

$$\rho(l_1, l_2, ..., l_m) = \prod_i \rho_i(l_i) \tag{1}$$

We ask the LLM for each $l_j$, to estimate the marginal proportion of solutions this holds for. As this may not add up to 1, we normalize the estimates to form a proper probability distribution. For simplicity, we focus in this work on settings where all solutions in the solution space are equally likely. As noted in Sec. 3.2, we encourage the LLM to propose balanced partitions. However, heuristic estimation allows us to support imbalanced partitions by reweighing the sampling to favor strata with more potential solutions. More details on prompting in App. F. In Fig. 2, the LLM determines the joint probabilities across two strata, the Mississippi River and the Missouri Compromise Line.

## 3.4 Probabilistic Prompting.

Post heuristic estimation, a set of strata are sampled from the joint probability distribution in Eqn. 1. This implicitly forms a *probabilistic prompt*, which specifies a distribution over concrete language model prompts. After a prompt is sampled, the LLM is then used to sample from within the stratum. Back to Fig. 2, East and South are sampled from the Mississippi and Missouri strata respectively, augmenting the final prompt with diverse specifications.

Formally, call $\vec{l}$ a stratum defined by choices of $l_{i,j}$ for each $P_i$ across all $i$. Call $Prompt$ a function that maps the stratum, $\vec{l}$, to a concrete prompt, $Prompt(\vec{l})$. The probabilities of the prompt distribution are defined by $Pr[Prompt(\vec{l})] = \rho(\vec{l})$. We then compute the probability of a solution as follows:

$$
\begin{aligned}
Pr[s] &= \sum_{\vec{l}} \Pr[\mathrm{Prompt}(\vec{l})] \; \Pr[s \mid \mathrm{Prompt}(\vec{l})] \\
&= \sum_{\vec{l}} \rho(\vec{l}) \; \Pr[s \mid \mathrm{Prompt}(\vec{l})].
\end{aligned}
\tag{2}
$$

The specific language model's next-token probabilities define $Pr[s \mid \mathrm{Prompt}(\vec{l})]$.

As the probabilistic prompt is in a human readable form, the user can inspect the properties and the proportions and modify it to adjust for unwanted bias or remove unwanted factors. For instance, when proposing English baby names, we may want the model to propose male vs female names equally often, even though there are more female than male baby names [2]. This interpretability and controllability is a major advantage of SimpleStrat in practice.

# 4 CoverageQA Dataset

## 4.1 Overview

We wish to evaluate generation diversity in settings where 1) user requests have more than one distinct correct answer, 2) and answers are equally likely, and 3) answers do not require hidden or implicit context to verify. These three features allow us to measure diversity quantitatively in terms of language model ability to represent the target distribution when resampled. The second condition makes the evaluation easier, reducing the problem to a measure of coverage over the solution space. Unfortunately, existing benchmarks discussed in Sec. 2 do not suffice. As such, we introduce CoverageQA for assessing the language model generation diversity. The dataset consists of two splits: CoverageQA-Curated, manually curated naturally underspecified questions, and CoverageQA-Wikipedia an auto-generated dataset of underspecified questions.

## 4.2 CoverageQA-Wikipedia Approach

To generate CoverageQA-Wikipedia, we leverage the Wikidata knowledge base which contains all relational mappings between entities and properties in Wikipedia. Our generation process starts with an initial item-property pairing and a constraint on the number of correct answers. We then perform a recursive search through Wikidata to find all sets of item-property constraints and their corresponding answers that meet our criteria. These constraints are subsequently transformed into natural language questions using GPT-4.

Consider an initial pairing of the Wikidata item "country" with the property "instance of". We might specify that we want between 20 and 40 valid answers. Our search would then yield a set of all constraints from the knowledge base that fit the initial conditions, such as "instance of country, located in Europe, uses Euro as currency". GPT-4 would convert this into a natural language question like "Name a country located in Europe that uses the Euro as its currency."

This approach has several advantages: 1) it allows us to create a diverse and extensive benchmark that can be easily updated with weekly updates to Wikidata, 2) it allows us to arbitrarily specify the size of the solution space as constraints can be added or removed to form; and 3) this process in principle can curate a large dataset with little manual effort or supervision. In the initial instantiation of CoverageQA-Wikipedia dataset, we publish 145 questions across 6 domains, corresponding to a different initial seed item-property pair. To ensure quality, we employ both automatic filters (e.g., excluding certain generic properties) and manual curation to remove redundant or unsuitable questions. This dataset can be substantially expanded as we only used 4 domains, but we leave this for future work. For details on the dataset breakdown and details on the question generation process, see Appendix A.

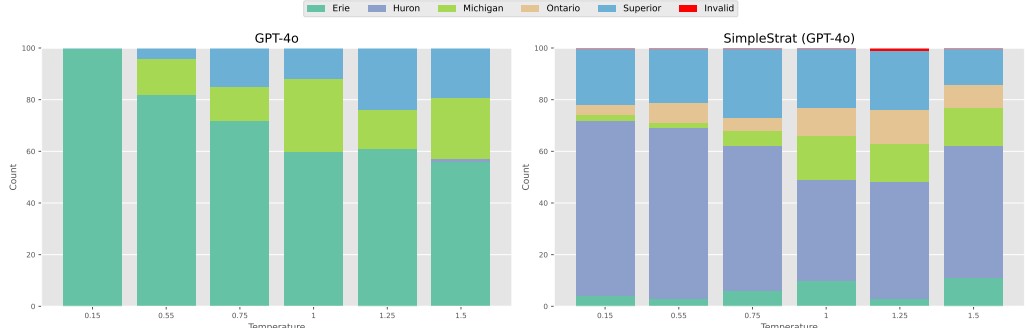

Figure 3: **Diversity scaled with temperature.** We show 100 resamples of "Name one Great Lake in the United States." On the left, we show the result of resampling GPT-4o 100 times per temperature. In contrast to SimpleStrat on the right, GPT-4o at temperature 1.5 still only samples Lake Huron once and never samples Lake Ontario. SimpleStrat improves the diversity across all temperatures.

## 5 Results

### 5.1 Measuring Diversity

We consider three measures of diversity. In the setting where we have access to all ground truth answers, we can measure *distributional diversity* when the model weights are available and *coverage diversity* when blackbox. Without a comprehensive list of valid solutions, we rely on embedding distance to measure *open-ended* diversity.

**Distributional diversity.** For models with accessible softmax next-token probabilities, we can compute the probability of each solution in the solution space. We then define distributional diversity as the distributional distance between the response distribution implied by the sampling process and logits and the ground-truth distribution derived from these probabilities. For our baseline, we prompt the models and directly compute $Pr[s|Prompt(\vec{l})]$ for each solution, $s$. This is simply the product of the individual next-token probabilities. For SimpleStrat, the probability involves the next-token probability conditioned on the probability the prompt is selected. Formally, the probability an answer is sampled by SimpleStrat can be computed based on Eqn.2. The next-token probability based response distribution $Pr[s|Prompt(\vec{l})]$ computed just as the baseline, and we do a sum weighted by the joint probabilities assigned in heuristic estimation. We assign the remaining probability density to an "Invalid" category to form a proper distribution. The probabilistic formulation allows us to easily compute the response distribution of SimpleStrat. Note that by design the ground-truth distribution for CoverageQA is uniform over valid solutions and zero elsewhere.

**Coverage diversity.** In setting where we do not have access to the next-token distribution, we evaluate diversity by resampling responses to CoverageQA 100 times per question. This allows us to empirically observe the diversity in the form of coverage. We call this coverage diversity. To measure coverage, we report the recall: unique valid solutions divided by total valid solutions on the reference solutions. This is not to be confused with a notion of recall where we might measure how many valid solutions a classifier recognizes as valid. To ensure this does not come at the cost of quality, we also show precision is not reduced. We show an ablation of just asking the model to propose criterion without 20 questions formulation in Fig. 4. We see little improvement from only allowing the model to propose a set of criterion and then applying the uniform sampling like AttrPrompt (Yu et al., 2023)

**Open-ended diversity.** For many compelling applications of diversity, a goal of diverse generation is to identify solutions not previously known to the user. In these open-ended settings, it's often the case that the space of valid solutions cannot be exhaustively enumerated. In this work, we consider creative writing prompts as an example of this setting. We ask the model to provide plot outlines to make comparison easier to decouple high level creative choices from stylistic choices. We measure diversity by sampling a pair of story plots outline based on the same prompt and measuring the cosine distance of the plots' embeddings. Unlike CoverageQA, writing a high quality plot is a more challenging, longer context task. As such, we assess the model's quality by spell-checking

---

[2]As reported in Wilson (2016), there are 18,993 unique names for girls and 13,959 for boys in 2015 report by Social Security Administration.

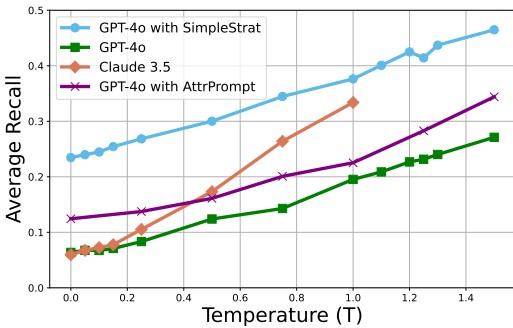

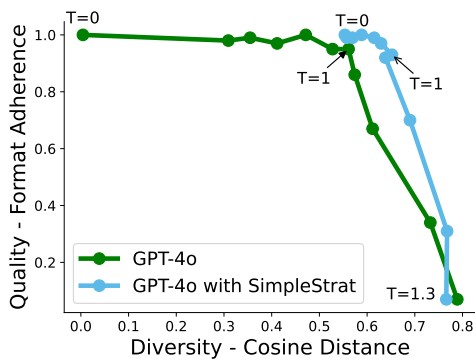

Figure 4: **Diversity measured with recall scaled with temperature.** SimpleStrat applied to GPT-4o shows the improved recall on CoverageQA compared to GPT-4o, AttrPrompt, Claude 3.5[3]. Recall indicates the percentage of ground truth questions observed after sampling 100 times. The benefit of SimpleStrat is especially pronounced at low temperatures, but the benefit is evident across all temperatures. Prior work, AttrPrompt, directly asks for partitions instead of our 20 questions style prompting, resulting in lower quality partitions.

Figure 5: **WritingPrompts Diversity.** On creative writing prompts, we generate pairs of plot outlines. We measure diversity with embedding cosine distance and quality by checking for format adherence. SimpleStrat especially improves the diversity at low temperatures achieving at T=0 the same diversity as T=1 for base GPT-4o.

for non-standard English words and adherence to a three act outline given in the prompt. These simple checks do not aspire to capture all aspects of quality but are sufficient to capture the quality degradation characteristic of high temperature. See App. H for more details.

## 5.2 Qualitative Example

Consider the question "Name one Great Lake in the United States." as shown in Fig. 3. We see that temperature scaling with GPT-4o results in a strong preference/bias for Lake Erie. This is certainly a correct continuation and under the language modeling objective should be incentivized. Increasing the temperature helps sample the next most likely candidate solutions more often. However, even when increasing the temperature past 1 there is still incomplete coverage over the solutions space. Specifically, Huron is only seen once out of 100 samples at 1.5 temperature, and Lake Ontario is never observed. This is undesirable if the data is used to propose candidate plans, generate test cases, or generate training data. Further, the model has a strong persistent preference for Lake Erie potentially leading to undesired biases in downstream use cases.

In Fig. 3, we observe a much more uniform distribution over valid solutions when using SimpleStrat. Notably, we observe full coverage over all 5 Great Lakes. At lower temperatures, there is still a preference for a single lake over the others, in this case Lake Huron. However, this is less pronounced at higher temperature showing the orthogonal benefit of SimpleStrat.

## 5.3 Coverage Diversity on Proprietary Models

We first assess coverage diversity, specifically, the model's ability to recall all the valid solutions upon resampling. This measure is clearly impacted by temperature as temperature zero or greedy decoding of LLMs leads to a single deterministic result. We compare the coverage diversity (recall) of SimpleStrat, GPT-4o, and Claude 3.5 Sonnet as a function of temperature. We sweep over temperatures from 0 to 1.5. SimpleStrat with GPT-4o leads to an improvement to recall across all temperatures as shown in Fig.4. Compared to prior work, AttrPrompt, SimpleStrat provides substantially more diversity when applied to the same underlying model (Yu et al., 2023). SimpleStrat scales well with temperature showing gains across all temperatures. The recall importantly does not come at the expense of quality as measured by precision as in App. D.

**Ablations.** We conduct two ablations to evaluate key design choices. First, we remove the **20 Questions** framing (**20Q Abl.**) and directly ask the model for partitions. Second, we test a **single-**

---

[3]Claude does not allow for temperatures above 1. GPT-5 was not studied because it does not allow for user-controlled temperature.

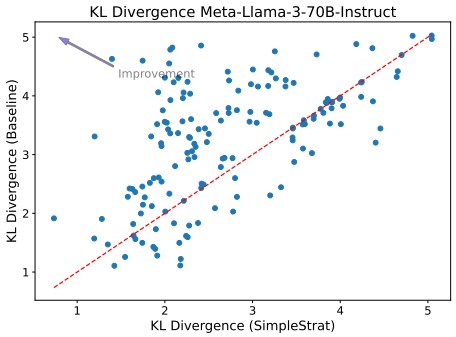

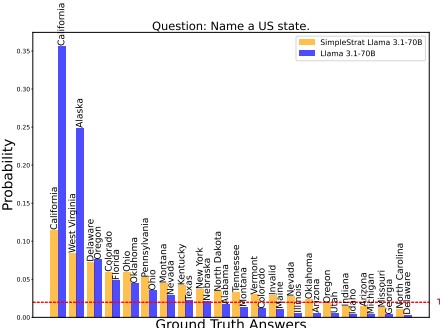

Figure 6: **KL divergence on individual question from CoverageQA Wikipedia.** For each question in CoverageQA, we compare the KL divergence from reference distribution of Llama-3 with and without SimpleStrat. Lower divergence indicates closer alignment with the desired uniform distribution, arrow indicates direction of maximum improvement from baseline

Figure 7: **Distributional Diversity Comparison.** We show the a posterior probability as defined by next-token probabilities for ground truth answers on Llama 3.1 70B. SimpleStrat provides meaningful improvement to the sampling distribution both for values previously overrepresented in the distribution and those previously underrepresented.

Table 1: Performance of Different Prompting Strategies across Temperature Settings (GPT-4o)

| Temp. | GPT-4o (std) | SimpleStrat (std) | 20Q Abl. (std) | Single Prompt Abl.(std) |
|---|---|---|---|---|
| 0 | 0.0646 (0.0011) | 0.2423 (0.0050) | 0.1215 (0.0021) | 0.0019 (0.0008) |
| 0.25 | 0.0849 (0.0016) | 0.2669 (0.0022) | 0.1405 (0.0028) | – |
| 0.5 | 0.1204 (0.0036) | 0.3017 (0.0043) | 0.1660 (0.0025) | – |
| 0.75 | 0.1497 (0.0046) | 0.3433 (0.0067) | 0.1821 (0.0336) | – |
| 1.0 | 0.1871 (0.0071) | 0.3884 (0.0094) | 0.2381 (0.0049) | 0.0272 (0.0014) |
| 1.25 | 0.2280 (0.0053) | 0.4250 (0.0062) | 0.2872 (0.0065) | – |
| 1.5 | 0.2676 (0.0059) | 0.4634 (0.0085) | 0.3304 (0.0104) | – |

**stage** version (**Single Prompt Abl.**) that combines all instructions into one prompt using the 20 Questions framing. Results in **Table 1** show that the single-prompt setup performs significantly worse than the baseline GPT-4o, suggesting that more tokens do not necessarily improve diversity and that our gains are not due to prompt complexity alone. This supports the intuition that probabilistic prompting—via "coin flips"—introduces beneficial randomness. The **20 Questions ablation** further confirms our 20 question framing provides better partitions.

## 5.4 Distributional diversity with Llama 3

We use Llama 3 models to measure distributional diversity by analyzing logits for all valid answers. This isn't feasible with GPT-4o or Claude 3.5 Sonnet, since estimating true probabilities would require heavy sampling. Although GPT models can output log probabilities, they report these only for the tokens in the observed generation trace, rather than for all possible continuations.

Across both 8B and 70B Llama models, SimpleStrat achieves an average reduction in KL divergence from uniform of 1.14 compared to the baseline on the curated CoverageQA dataset. For the general CoverageQA dataset, the reduction is 0.36. These results indicate that SimpleStrat produces a response distribution closer to the ground truth distribution than the baseline method.

Additionally, we analyze per-question KL divergence with the scatter plot in Fig 6. It shows KL divergence values for SimpleStrat (y-axis) versus the baseline (x-axis) for each question in the CoverageQA Wikipedia dataset. Points above the diagonal line represent questions where SimpleStrat outperforms the baseline by yielding a lower KL divergence. Points tend to fall near or above this line, indicating SimpleStrat produces more uniform samples on CoverageQA.

As shown in Fig. 7, Llama 3.1's base distribution is heavily biased toward its preferred answer (e.g., "California," as shown also in Fig. 1). Thus, it is not surprising that we observed little diversity when simply increasing temperature. In contrast, SimpleStrat provides a much more uniform distribution. The overrepresented solutions are adjusted to be lower and the underrepresented solutions are adjusted to be higher. For more examples, see App. G.

Table 2: We assess distributional diversity as measured by KL Divergence. Smaller KL divergence is closer to uniform. We see improvement to distributional diversity for both 8B and 70B as well as Llama-3 and Llama-3.1.

| Model | CoverageQA-Curated | | CoverageQA-Wiki | |
|---|---|---|---|---|
| | Baseline | SimpleStrat | Baseline | SimpleStrat |
| Llama-3-8B-Instruct | 2.78 | **1.74** | 2.75 | **2.47** |
| Llama-3.1-8B-Instruct | 2.47 | **1.19** | 2.60 | **2.39** |
| Llama-3-70-Instruct | 3.24 | **2.17** | 3.28 | **2.73** |
| Llama-3.1-70-Instruct | 2.70 | **1.54** | 2.78 | **2.38** |

## 5.5 Open-ended Diversity.

Fig. 5 shows SimpleStrat shifts the curve to the right, affording diversity while maintaining the same quality. We provide a coarse grain measure of diversity by measuring the proportion of plots from the 100 random prompts taken from WritingPrompts that do not have formatting errors or more than 20% of words outside of the English dictionary. As we are asking for an outline, format errors we check for include proper monotonic numbering and providing the number of requested chapters to the story. Notably, at temperature zero, we achieve the same diversity as temperature scaling to temperature 1. This suggests we get diversity for free without sacrificing quality.

## 6  Limitations and Future Work

While SimpleStrat demonstrates empirical gains, its effectiveness depends on the model's ability to identify meaningful axes in auto-stratification and estimate accurate joint probabilities. As LLMs improve in forecasting and external data integration, we expect these estimates to become more reliable. Our prototype focuses on the model's intrinsic capabilities, but potential biases—such as those related to race or gender—may influence stratification and estimation. For critical applications, the probabilistic prompt distribution should therefore be carefully reviewed. Finally, because CoverageQA consists of short responses, evaluation is simplified; however, we anticipate SimpleStrat will have the greatest impact in low-temperature, multi-step reasoning tasks (Zhang et al., 2024).

As research on learning beyond demonstrations and reinforcement learning accelerates, methods that promote diversity—such as SimpleStrat—are poised to become central to discovering novel solutions, strategies, and ideas. Wherever diversity is currently achieved through temperature scaling, SimpleStrat provides a more semantically grounded alternative. Task-proposing agents like InSTA and Explorer, for instance, could leverage SimpleStrat to explore websites more effectively and generate a broader range of trajectories for web agent training (Trabucco et al., 2025; Pahuja et al., 2025). Similarly, AlphaEvolve employs evolutionary strategies to tackle optimization problems and references "stochastic formatting" as a way to introduce variance into prompts—likely an early form of probabilistic prompting akin to the approach described in this work (Novikov et al., 2025).

## 7  Conclusion

In this paper, we propose SimpleStrat which offers an innovative alternative by leveraging the LLM itself to partition the solution space into distinct strata. We call this process *auto-stratification*. Specifically, we reframe the stratification problem to the imperfect information game of *20 questions* and show that this framing produces strata that are both balanced and orthogonal. At inference time, a random stratum is selected, and a sample is drawn from within that stratum. This method achieves greater diversity while maintaining quality, unlike simply increasing temperature.

To quantitatively measure diversity, we introduced the CoverageQA dataset, which consists of underspecified questions with multiple equally valid answers. We measure diversity with three metrics: for open-source models, we measure distributional difference with KL Divergence and for proprietary models, we measure coverage over the set of ground-truth solutions. In the open-ended setting without access to the ground truth distribution, we rely on distance in embedding space to measure diversity. Our rigorous evaluation on both proprietary and open-source LLMs demonstrated that SimpleStrat achieves significantly higher recall and produces answer distributions closer to uniform compared to traditional temperature-based sampling methods.

**Acknowledgement**

We thank Anastasios Angelopoulos, Jacob Steinhardt, Brandon Trubucco, Kevin Yang, Alan Zhu, and Parth Asawa for their insightful discussion. This work was supported in part by DARPA Contract FA8750-23-C-0080 (ANSR), by Nissan and Toyota under the iCyPhy center, and by Sky Computing Lab which is supported by gifts from Accenture, AMD, Anyscale, Broadcom, Cisco, Google, IBM, Intel, Intesa Sanpaolo, Lambda, Lightspeed, Mibura, Microsoft, NVIDIA, Samsung SDS, and SAP.

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

# A    CoverageQA Dataset

**Generation Procedure**

To generate the questions, we manually came up with initial item and property pairings to run the recursive search. We constrain the recursive search to yield between 20-40 possible answers to keep the questions within common and relevant categories. We found that with fewer than 20 answers, the questions become too obvious, while with more than 40, they tend to get too specific and stray from general knowledge. The recursive search first finds all items that satisfy the initial conditions, then iteratively adds properties in steps until either the maximum depth (number of constraints) is reached or the number of answers falls outside the desired range. We blacklist properties that are detrimental to high-quality question generation, such as an item's presence in a specific database, numeric properties like population, and properties that introduce high ambiguity. We then manually evaluate the generated conditions and answers to ensure they meet our criteria. With an appropriate initial condition, one query can generate hundreds of valid constraints that can later be turned into questions. Finally, we use GPT-4 to convert these constraints into natural language.

# B    Results on Curated CoverageQA

We manually curate questions with known solutions sets such as NFL teams and USA state capitals. This bank serve as a split to validate results on less synthetic sources that are more common knowledge. We see even stronger performance improvement on this set in Fig. 8

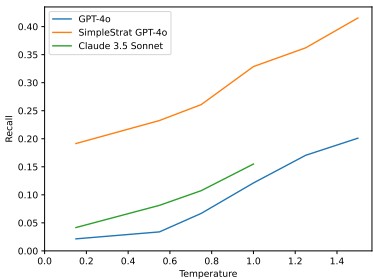

Figure 8: We see even stronger improvements on the manually curated CoverageQA set.

# C    Experimental Setup

For our evaluation, we use proprietary models, gpt-4o-2024-08-06 and claude-3.5-sonnet-20240620. We use open-source models from the Llama 3 and 3.1 families. The inference of these models were run on 8 A100-80GB GPUs. CoverageQA Wikipedia is based on the snapshot from 07-03-2024. For text embeddings, we use 3rd generation embedding from OpenAI.

Table 3: CoverageQA Domains

| Domain | Question Count | Average Number of Answers |
|---|---|---|
| General Knowledge (Curated) | 10 | 64.1 |
| US National Parks | 5 | 11 |
| Geography Questions | 74 | 27.5 |
| Periodic Table Elements | 11 | 24.2 |
| Physic Nobel Laureates | 31 | 16.8 |
| Famous Athletes | 18 | 9 |
| Musical Instruments | 6 | 10 |
| Total | 155 | 24.1 |

# D   Precision

We show precision in Fig. 9 to emphasize that the precision does not change substantially as a result of our method. Precision is calculated over the set of 100 attempts how many are in the ground truth. Recall as mentioned is calculated as how many unique ground truth solutions were observed in the 100 attempts. The reduced precision can be attributed to cases where the Heuristic Estimation is ineffective. This can lead to settings where a stratum has no valid solution. Because the constraints are added as additional conditioning, the model now has a competing objective to obey the constraints vs the original instruction. As such, infeasible requests lead to best effort solutions that are incorrect.

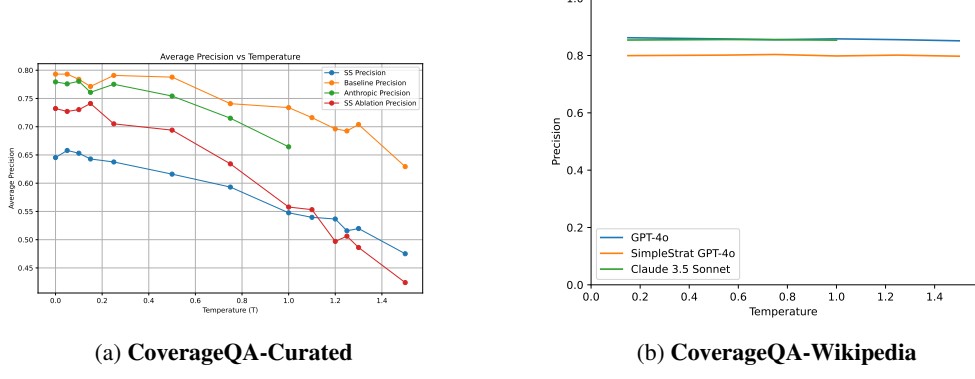

(a) **CoverageQA-Curated**                    (b) **CoverageQA-Wikipedia**

Figure 9: **Precision scaled with temperature.** There is a fixed minor reduction of 10% to precision when using SimpleStrat. This indicates the improved diversity does not come at a significant cost to precision.

# E   Auto-stratification Prompt

We provide the full prompt in Tbl. 4. To improve prompt adherence, we provide one in context example in the form of one simulated round of multi-turn conversation, i.e. we provide an example set of reasoning following the template.

---

**System Prompt:**
You're a helpful brainstorming assistant that is careful to consider all factors to a problem.

---

**User:**
I am tasked with the following request:
*% User Request*
Help me brainstorm how to respond to the user request by providing a list of True/False properties the solution may or may not have. Use the following step-by-step to come up with good properties:

1. If you were playing 20 questions, what's a good first question to ask that would split the possibilities in half?

   List at least 5 questions and their corresponding properties.

   Question: <Description>

2. Rewrite each question as a True/False property that's true for one half and false for the other.

   Question: <Description>

   True/False Property: <Property Description>

3. For each property, come up with an example that would satisfy the property.

   Property: <Description>

   Example: <Description>

   Is it a valid answer to the user's request? <Yes/No>

4. For each property, come up with an example that would not satisfy the property.

   Property: <Description>

   Example: <Description>

   Is it a valid answer to the user's request? <Yes/No>

5. Does the property mention a candidate answer in it?

   Property: <Description>

   Does the property mention a candidate answer in it? <Yes/No>

6. For each property, list whether we should include it or not in the final list of properties. Do not include ones where an example from above is not valid or if it mentions a candidate answer in it.

   Property: <Description>

   Include in final list? <Yes/No>

Final List of True/False Properties:

1. <Property Description 1>
2. <Property Description 2>

Ensure all properties are listed are sentences that are either True or False

---

Table 4: Full prompt for Auto-stratification.

---

**System Prompt:**
You are an expert superforecaster, familiar with the work of Tetlock and others. Your mission is to generate accurate predictions for forecasting questions. Aggregate the information provided by the user. Make sure to give detailed reasoning.

---

**User:**
I am tasked to estimate the probability that a random solution to "*User Request*" has the following property "*Partitioning Property*"
Instructions:

1. Provide at least 3 reasons why the answer might be no.

   { Insert your thoughts }

2. Provide at least 3 reasons why the answer might be yes.

   { Insert your thoughts }

3. Rate the strength of each of the reasons given in the last two responses. Think like a superforecaster (e.g. Nate Silver).

   { Insert your rating of the strength of each reason }

4. Aggregate your considerations.

   { Insert your aggregated considerations }

5. Output your answer (a number between 0 and 1) with an asterisk at the beginning and end of the decimal.

   { Insert your answer }

---

Table 5: Prompt for Partition-specific Heuristic Estimation.

# F Heuristic Estimation Prompt

We first take each partition function from auto-stratification and estimate a starting probability with the prompt in Table 5. This prompt is heavily inspired by Halawi et al. (2024). We then collect all the proportions and pass it through a final Heuristic Estimation prompt to remove redundant properties (negations for instance) and give the model a chance to correct any incorrect probabilities. See Table 6 for full prompt. Finally, we ask the model to select at most 3.

Note that for performance reasons, we estimate the marginal probabilities and make a simplifying assumption of independence. This is not strictly true if one partition function is the negation of the other. This leads potential stratum assigned positive probability but actually the stratum has no solutions. Otherwise, there would be $2^{\text{\# of Partition Functions}}$ strata to estimate probabilities of. Further, LLMs seem less reliable when asked to estimate fine-grained probabilities, whereas most marginal probabilities are by design close to 0.5.

Formally, if $P = \neg Q$, the the stratum $P \wedge Q$ has zero probability, even though we assumed it to be $Pr[P] * Pr[Q]$. We handle approximation error in estimating the true prompt distribution by allowing the model to reply "Invalid" to trigger a resample. With this adjustment, the probabilistic prompt distribution is maintained for this extreme case. This correction however does not ameliorate potential issues with

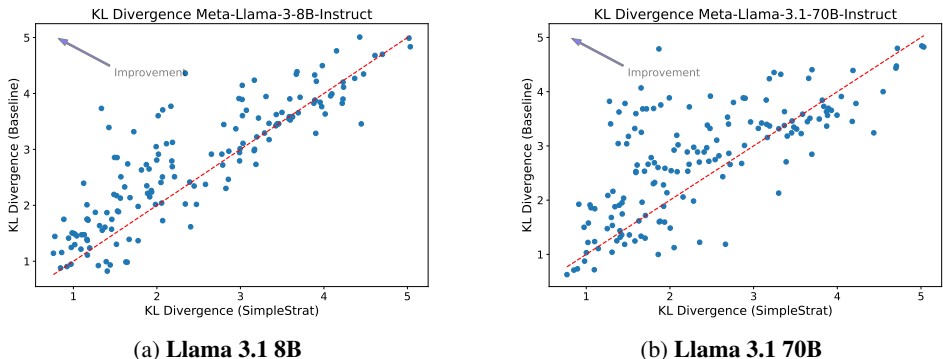

(a) **Llama 3.1 8B**      (b) **Llama 3.1 70B**

Figure 10: KL divergence from uniform for Baseline vs SimpleStrat on CoverageQA Wikipedia. Lower divergence indicates closer alignment with the desired uniform distribution, arrow indicates direction of maximum improvement from baseline

---

**System Prompt:**
You are an expert superforecaster, familiar with the work of Tetlock and others. Your mission is to generate accurate predictions for forecasting questions. Aggregate the information provided by the user. Make sure to give detailed reasoning.

---

**User:**
I'm playing a game where my friend has been tasked to:
"*User Request*"
I have the following Y/N statements I can ask my friend. I have probabilities that I think it's true: % Insert numbered list of partitions and proportions.
Instructions:

1. For each Y/N statement, is it redundant with another statement?

   Y/N statement: <description>

   Is redundant? <Y/N: Explanation>

2. Are any of the probabilities in accurate? If it's sufficiently accurate just report back the same value.

   Y/N statement: <Description>

   Is accurate? <Y/N: Explanation>

   Probability: <Probability>

3. Pick at most three statements that are least redundant and pair well together. Prefer ones that are closest to 50% for most information.

Final List of True/False Properties:

1. <Y/N Properties> :: <Probability>

2. <Y/N Properties> :: <Probability>

---

Table 6: Prompt for Final Heuristic Estimation.

# G    Additional Plots: Distributional Analysis with Llama

We provide additional examples in Fig 15 and scatter plots for Llama 3 in Fig 10a, Fig 10b, and Fig 11.

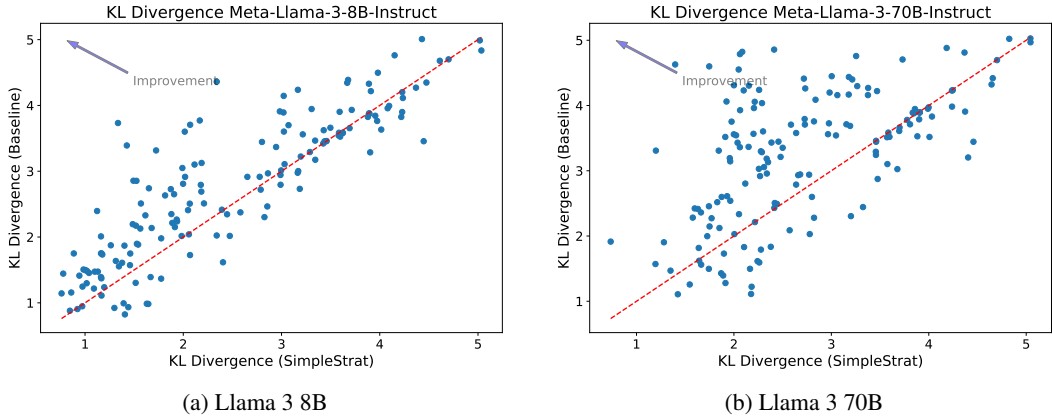

(a) Llama 3 8B

(b) Llama 3 70B

Figure 11: KL divergence from uniform for Baseline vs SimpleStrat on CoverageQA Wikipedia. Additional plots for Llama 3 8B and 70B models

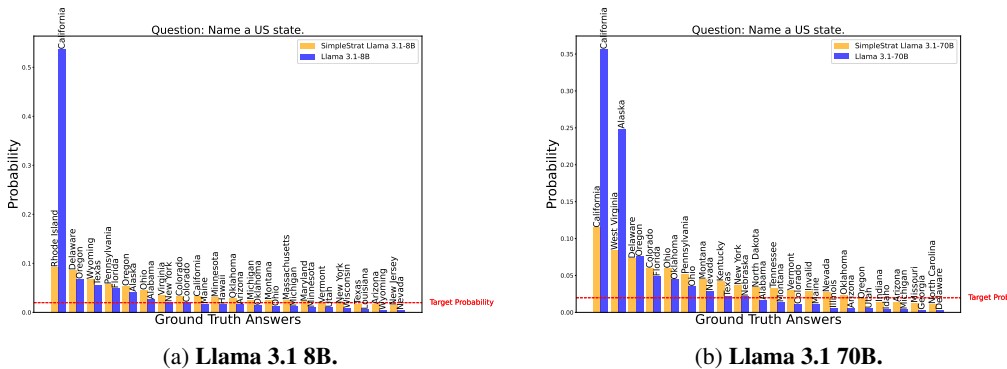

(a) **Llama 3.1 8B.**

(b) **Llama 3.1 70B.**

Figure 12: **Distributional Diversity Comparison.** We show the response probability as defined by next-token-probabilities for the top 20 ground truth answers on Llama 3.1. For both 8B and 70B, SimpleStrat provides meaningful improvement to the response distribution both for values previously over-represented in the distribution and those previously underrepresented.

# H    Judging Plot Outlines

For this task, we ask the model to generate outlines guided by the format shown in Table 7. Notably, we ask for three acts to prevent the outlines from getting too long. To check for validity, we use pyspellcheck to assess if there are over 20% words not in the English dictionary and ensure that there are three acts and a THE  END to finish the story. The last condition protects against the case where the model rambles on incoherently and produces what is definitely not a sensible outline. With spelling alone, we can already see the degradation of the model due to temperature (Fig. 13). Finally, we see formatting is not affected by SimpleStrat and largely tracks the temperature used to sample from the language model (Fig. 14).

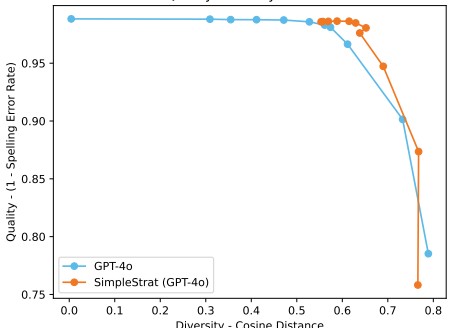

Figure 13: **WritingPrompts Diversity.** On creative writing prompts, we generate pairs of plot outlines. We measure diversity with embedding cosine distance and quality proxied by spelling. SimpleStrat especially improves the diversity at low temperatures achieving at T=0 the same diversity as T=1 for base GPT-4o.

Figure 14: **Impact of temperature on formatting (including spelling).** It's interesting to see that the impact of SimpleStrat is negligible compared to temperature. We see the SimpleStrat closely follows the expected quality degradation based on the temperature of the model generating the final outline.

**System Prompt:**
Format:

**Title**: <TITLE>

**Setting**: <SETTING>

**Characters**: <CHARACTERS>

**Act 1:** <ACT 1 TITLE>
1. <Content>
2. <Content>
3. <Content>
...

**Act 2:** <ACT 2 TITLE>
1. <Content>
2. <Content>
3. <Content>
...

**Act 3:** <ACT 3 TITLE>
1. <Content>
2. <Content>
3. <Content>
...

THE END

**User:**
Write a 3 part story outline based on the following prompt:
"*User Request*"

Table 7: Prompt for Plot Outlines.

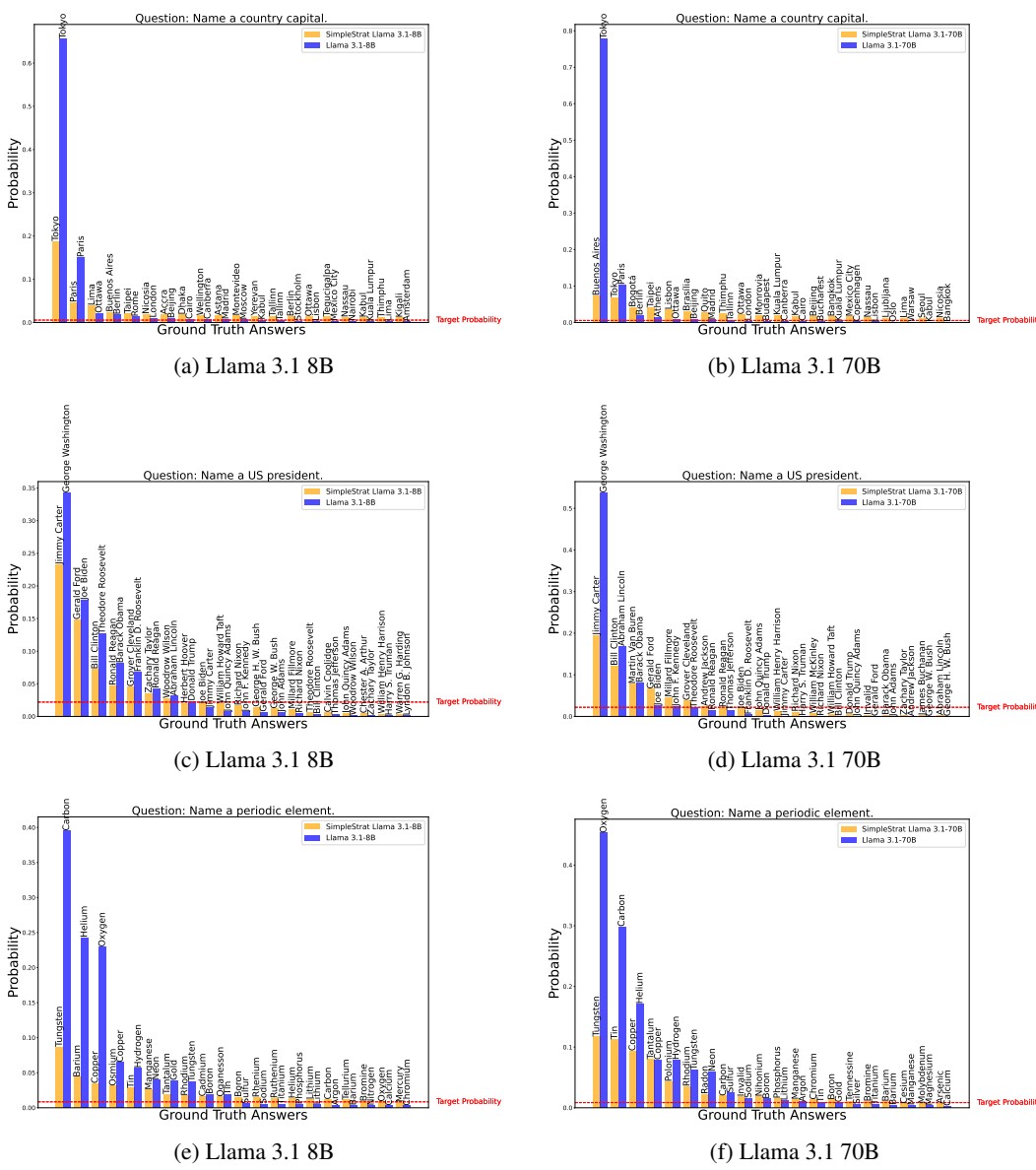

Figure 15: **Baseline vs SimpleStrat Probability Distributions** This figure shows the answer distributions for 4 additional questions from CoverageQA curated. Each row represents a different question, showing distributions for Llama 3.1 8B and 70B.

