# OpenReview forum: "SimpleStrat: Diversifying Language Model Generation with Stratification"
_NeurIPS.cc/2025/Conference — NeurIPS 2025 poster_

### Official Review · Reviewer_UC4f · 2025-07-02

**Clarity:** 3
**Significance:** 2
**Originality:** 2
**Rating:** 4
**Confidence:** 2

**Summary:**

The authors introduce SimpleStrat, a new method designed to enhance diversity in LLMs. SimpleStrat uses stratified sampling guided by auto-generated criteria (auto-stratification) from the LLM itself. To evaluate the quality of the sampling method the authors introduce CoverageQA, a benchmark to evaluate response diversity. SimpleStrat performs well on the introduced benchmarks.

**Questions:**

Do you think it is possible to do something similar but without the need for computing stratification prompts?

**Ethical Concerns:**

["NO or VERY MINOR ethics concerns only"]

**Final Justification:**

I have read the rebuttal and my concerns remain largely unchanged. Also upon revisiting the work I have some concerns about whether it meets the high bar of NeurIPS. I will leave my score unchanged.

**Limitations:**

yes

**Quality:**

3

**Strengths And Weaknesses:**

Strengths:
- The paper addresses an important and promising area—improving sampling methods in LLMs—which remains relatively underexplored compared to, e.g., diffusion models where this is a very active area of research.
- The methodology is intuitive and well-explained, featuring detailed examples that illustrate how SimpleStrat operates. The newly introduced benchmark CoverageQA seems like a step into the right direction for assessing the quality of such methods.
- The empirical evaluation, while mostly limited to CoverageQA shows promising results across the considered measures of diversity (KL divergence, recall, embedding distances).

Weaknesses:
- The effectiveness of SimpleStrat heavily relies on the LLM's ability to propose meaningful stratification criteria through self-prompting. Poor initial stratification could reduce potential benefits significantly. Relying exclusively on self-prompting for stratification might miss opportunities for more sophisticated stratification methods—perhaps leveraging neuron activations or intermediate representations could yield more effective stratifications.
- Computational overhead may be substantial due to multiple stages (auto-stratification and heuristic estimation), possibly limiting applicability in resource-constrained or latency-sensitive scenarios.
- While useful, the CoverageQA dataset’s heavy reliance on automatically generated constraints from Wikidata could lead to overly synthetic or unintuitive scenarios, potentially limiting real-world applicability. It would be nice to validate the dataset via labelling a small set with human annotators. The same would be nice to assess the quality of the sampling method. Does sampling with this method improve the scores on something like LMArena?

---

> ### Author Rebuttal · Authors · 2025-07-29
>
> We appreciate the thoughtful review and the constructive feedback! We hope the following clarifications and experiments address your questions.
>
> ### Limitations of Self Prompting:
> It’s indeed the case that the approach is sensitive to the quality of the stratification. In fact, our ablation in Figure 4 indicates the 20-questions prompting-approach is crucial to the quality of the stratification and subsequently the resulting diversity.
>
> It’s certainly true that stratification can be discovered with other techniques. We are familiar with (Hu et al): https://arxiv.org/pdf/2505.05145 which analyzes the contribution of the attention heads to the final prediction, but the semantic analysis is limited to simple tasks. It’s not immediately obvious if the head-wise distinction would always lead to semantically interpretable strata (possibly better with SAE). Is this what you mean by leveraging activations? We would be curious to understand better how neuron activation and intermediate representations can be used to identify semantic strata as future work.
>
> We added an additional ablation to emphasize the importance of quality partitions by only changing the 20 questions:
>
> ##### CoverageQA Results with std and additional Ablations -- Original Figure 4
>
> | Temperature | Baseline GPT-4o (std) | SimpleStrat with GPT-4o (std) | AttrPrompt with GPT-4o (std) | SimpleStrat w/o 20-questions| SimpleStrat Single Prompt with GPT-4o (std)|
> |-------------|-----------------------|-------------------------------|---------------------------------------|--|-|
> |0 | 0.0646 (0.0011) | 0.2423 (0.0050) | 0.1244 (0.0006) | 0.1215 (0.0021)| 0.0019 (0.0008) |
> |0.25 | 0.0849 (0.0016) | 0.2669 (0.0022) | 0.1389 (0.0021) | 0.1405 (0.0028)| - |
> |0.5 | 0.1204 (0.0036) | 0.3017 (0.0043) | 0.1619 (0.0027) | 0.1660 (0.0025)| - |
> |0.75 | 0.1497 (0.0046) | 0.3433 (0.0067) | 0.1925 (0.0061) | 0.1821 (0.0336)| - |
> |1 | 0.1871 (0.0071) | 0.3884 (0.0094) | 0.2265 (0.0035) | 0.2381 (0.0049)| 0.0272 (0.0014)|
> |1.25 | 0.2280 (0.0053) | 0.4250 (0.0062) | 0.2824 (0.0043) | 0.2872 (0.0065)| - |
> |1.5 | 0.2676 (0.0059) | 0.4634 (0.0085) | 0.3360 (0.0042) | 0.3304 (0.0104)| - |
>
> We note that in addition to relying on quality partitions. The inclusion of external variance introduced by sampling constraints is also critical. We see SinglePrompt approach where we append the prompts together and allow the model in prompt space to select the positive or negative constraint lead to signification regression even compared to baseline GPT-4o. Noting that the language model has been trained to ignore random irrelevant tokens. The key to self-stratification is producing token efficient constraints with outsized impact on model's diversity. Critically, these small number of constraints allowing the system to inject external randomness to token space giving $2^n$ candidate prompts.
>
> ### Computational overhead:
>
> As articulated in section 3.1, we advocate for caching of AutoStratification and Heuristic Estimation per prompt. Effectively, once the stratification has been discovered, it can be reused and the excess token overhead viewed as amortized cost across many rounds of resampling. For a cached prompt, only a small overhead of ~20 tokens per constraint (3-5 constraints per problem on average) is needed for each query in Probabilistic Prompting.
>
> Under extreme latency conditions, the model can be assumed to be small (1B-8B) with latency below 10ms to the first token. This should render the minimal per query overhead to be negligible. In highly-resource constrained and latency-sensitive settings, we can go further to optimize latency by precomputing stratification for common prompts.
>
> ### CoverageQA realism:
> We agree there is some risk of a sim-to-real gap between the Wikipedia CoverageQA questions/answers and more natural human generated questions/answers. We have taken several measures to address this:
>
> 1. As discussed in line 247, we manually curate the answer set to ensure the answers are reasonable. Although the heavy-lifting has been done by the WikiData community. This manual check was crucial in both removing poorly phrased or ambiguous questions but more importantly identifying missing answers. For instance, certain noble gasses were missing from the WikiData queries. We are also careful to not over punish the model by enumerating possible ways of expressing the same answer – for instance countries (e.g. "United Arab Emirates","UAE", and "Emirates") as well as for names (e.g. "J. Robert Oppenheimer", "Robert Oppenheimer", "Oppenheimer", and "J. R. Oppenheimer").
>
> 2. We separately show evaluation on CoverageQA-curated in Appendix B. This set represents a small set of 15 questions generated by the authors: e.g. give me a NFC East team. Although at a smaller scale, this evaluation gives us some confidence the wiki generated questions are representative of real questions. SimpleStrat shows similar improvement in that setting.
>
> 3. WritingPrompts experiments show diversity improvement on prompts purely human generated on Reddit. Although diversity is measured differently given the open-endedness of the prompts, this gives us confidence that SimpleStrat discovers productive stratification.
>
> Although these measures have been taken, we agree human annotation and pair-wise comparison like LMArena can be added to further improve the quality of the CoverageQA dataset and potentially broaden the scope of questions to more subjective domains.

---

> > ### Author Response · Authors · 2025-08-04
> >
> > We hope our rebuttal has clarified the questions raised in the review. If you have any further questions or concerns, we’re happy to follow up.

---

> > ### Comment · Reviewer_UC4f · 2025-08-05
> > **rebuttal response**
> >
> > I have read the rebuttal and my concerns remain largely unchanged: creating the 20 stratifications seems unpractical in many situations and the evaluation is on a synthetic toy-task. Also upon revisiting the work I have some concerns about whether it meets the high bar of NeurIPS. I will leave my score unchanged.

---

> ### Author Response · Authors · 2025-08-05
>
> Thanks for reading our rebuttal. As a minor clarification, we do not create 20 stratification but rather prompt (Appendix E) the model to propose 5 candidate first questions to ask in "20 questions" the game. This game requires players to try to narrow down the "hidden" item within 20 questions. As such the questions at any point of the game should eliminate half of the solutions space.
>
> Measuring diversity is a difficult open problem. As such, we believe the settings we introduce of *creative writing* and *questions answering in the setting of multiple answers* is a good balance of complexity while allowing for concrete measures of diversity. This said, we are open to evaluating on additional settings if you have suggestions.
>
> We appreciate your overall positive reception and agree that diversity in language modeling "remains relatively underexplored compared to, e.g., diffusion models where this is a very active area of research." We are excited to see where this research direction leads.

---

### Official Review · Reviewer_ucfG · 2025-07-03

**Clarity:** 3
**Significance:** 3
**Originality:** 3
**Rating:** 5
**Confidence:** 3

**Summary:**

The paper proposes an approach called SimpleStrat using the language model itself to partition the answer/solution space into plausible regions from where sampling is performed. The authors develop a dataset of underspecified questions called  CoverageQA with multiple equally plausible answers. The paper also proposes to use KL divergence for measuring the resampling diversity between the output and true distributions. The paper reports better recall and average reduction in KL divergence when compared with widely used LLMs.

**Questions:**

1. The authors should mention why they chose certain LLMs over others for their experiments.

**Ethical Concerns:**

["NO or VERY MINOR ethics concerns only"]

**Final Justification:**

I have already reviewed the author rebuttal and I am satisfied by the response to the issues addressed by the authors.

**Limitations:**

Yes

**Quality:**

3

**Strengths And Weaknesses:**

SimpleStrat is a training-free approach to improve diversity with auto-stratification, heuristic estimation,  and probabilistic prompting by controlling the randomness in prompts. The workflow of the approach is very intuitive with focus on tackling the pain points. All the steps are clearly defined with examples. The comparison with temperature as a factor is insightful.

It is not clear why only Llama 3.1 models are considered for the experiments with KL divergence. The authors should mention the selection criteria for the chosen LLMs.

---

> ### Author Rebuttal · Authors · 2025-07-29
>
> We appreciate the thoughtful and enthusiastic review.
>
> ### Temperature comparison:
>
> We also found the temperature scaling comparisons “insightful” and emphasize the importance of finding general purpose approaches to improving output variability orthogonal to temperature scaling which trades off prompt adherence and diversity.
>
> ### Choice of Language Models:
>
> *Open-source models.* At the time of selection, we chose Llama 3.1 as it was the state-of-the-art open-weight model. We will include Qwen-3 as well in the final revision as it has grown in popularity and capability. We do not anticipate significant differences in results as new open-source models get released since the proprietary models also show the same limitation. Particularly, we anticipate mode-collapse in instruction-tuned models may become even more severe in reasoning models where models have been trained on objective single-answer math problems. We leave analysis of reasoning focused models as future work.
>
> *Proprietary models.* We chose OpenAI’s 4o models as and claude-3.5-sonnet as they demonstrate the key general domain capabilities characteristic of large language models, while remaining affordable for extensive resampling experiments. As models are becoming cheaper to experiment with, we are happy to include more models in the evaluation if desired.
>
> ### Intuitive presentation
>
> We are glad the reviewer finds the approach intuitive with the carefully chosen examples we used to illustrate the approach. We believe the intuitive simplicity of SimpleStrat makes it easy to incorporate in other settings.

---

> > ### Author Response · Authors · 2025-08-04
> >
> > Thanks for the enthusiastic review! If you have any further questions or concerns, we’re happy to follow up.

---

> > ### Comment · Reviewer_ucfG · 2025-08-05
> > **Response to Author Rebuttal**
> >
> > Thanks for addressing my comments. I am satisfied by the answers by the authors. If the responses are added to the final version of the paper, it can be a valuable addition in the field of mathematical reasoning by LLMs.

---

> > > ### Author Response · Authors · 2025-08-05
> > >
> > > Thanks! We appreciate your feedback and will certainly include a detailed discussion of why we selected the models we did to the final version of the paper. As new models are getting released (gpt-oss today), we will also include them in the KL-Divergence analysis.
> > >
> > > We agree that the KL-divergence comparison between the true distribution and observed distribution provides a practical and unique lens to understand and measure diversity.

---

### Official Review · Reviewer_VKJt · 2025-07-07

**Clarity:** 2
**Significance:** 2
**Originality:** 2
**Rating:** 3
**Confidence:** 4

**Summary:**

This work aims to improve the sampling diversity of LLMs for tasks that are open-ended or have multiple valid solutions. To this end, the authors introduce CoverageQA, a dataset of underspecified questions with multiple equally plausible answers. A multi-stage prompting method, SimpleStrat, is also proposed to improve the sampling diversity. The empirical experiments demonstrate the effectiveness of the proposed prompting method across multiple settings.

**Questions:**

1. What is “Anthropic Recall” in Figure 4? It does not appear to be mentioned anywhere in the main text.

2. How exactly is “SS Ablation” different from “SS”? The description is somewhat vague.

**Ethical Concerns:**

["NO or VERY MINOR ethics concerns only"]

**Final Justification:**

As discussed in my response to the authors' rebuttal, I believe that more thorough empirical comparisons with related methods/baselines for improving sampling diversity should be considered to better demonstrate the significance of the proposed approach. Therefore, I maintain my original assessment.

**Limitations:**

Yes

**Quality:**

2

**Strengths And Weaknesses:**

Strengths:

1. The studied problem, improving the sampling diversity of LLMs, is important and has practical value.

2. The introduced dataset, CoverageQA, could be a useful resource for studying the diversity and coverage of LLM generations.

3. The proposed prompting method, SimpleStrat, outperforms baselines across various evaluation settings.

Weaknesses:

1. Stronger baselines from prior work should be included to better validate the advantages of the proposed method, such as [1][2][3]. In particular, no baselines aimed at increasing sampling diversity appear to be compared.

2. More thorough ablation studies should be conducted to better understand the role of each component in the proposed method, SimpleStrat, which would help clarify the impact of each stage. For example, a single-stage prompting variant can be compared to the original multi-stage version.

References

[1] Zhu, Wenhong, et al. "Improving Open-Ended Text Generation via Adaptive Decoding." ICML 2024.

[2] Adaptive Contrastive Search: Uncertainty-Guided Decoding for Open-Ended Text Generation (Garces Arias et al., EMNLP Findings 2024)

[3] Meyerson, Elliot, et al. "Language model crossover: Variation through few-shot prompting." ACM Transactions on Evolutionary Learning 4.4 (2024): 1-40.

---

> ### Author Rebuttal · Authors · 2025-07-29
>
> We appreciate the thoughtful review and the constructive feedback! We hope the following clarifications and experiments address your questions.
>
> ### Figure 4 Legend:
> `Anthropic` stands for Claude-3.5 sonnet. This is in contrast to the `Baseline` which stands for GPT-4o which is a more direct comparison since the instantiation of SimpleStrat we implement uses GPT-4o. SS stands for SimpleStrat, SS Ablation stands for the Ablation described above where we replace the 20-questions approach to AutoStratification with a direct prompt asking for partitions. Thanks for raising bringing this confusion to our attention; we will rename the manuscript's legend to be clear and consistent with the main text's names.
>
> ### Ablation:
> The existing ablation coined `SS Ablation` in the main result in Figure 4 constitutes replacing both using a direct prompt instead of the 20-questions inspired prompting in AutoStratification and heuristic estimation is replaced with uniform sampling instead of heuristic estimation.
>
> In addition to the ablations provided in the original manuscript, we add two additional ablations: as suggested, we consider the setting where we provide all the instructions to the model asking it to generate both the partitions and the sampling probabilities in one step. Second, to analyze the importance of the 20-questions approach to AS, we substitute only that prompt with a simple direct prompt asking for partitions.
>
> | Temperature | Baseline GPT-4o (std) | SimpleStrat with GPT-4o (std) | AttrPrompt with GPT-4o (std) | SimpleStrat w/o 20-questions| SimpleStrat Single Prompt with GPT-4o (std)|
> |-------------|-----------------------|-------------------------------|---------------------------------------|--|-|
> |0 | 0.0646 (0.0011) | 0.2423 (0.0050) | 0.1244 (0.0006) | 0.1215 (0.0021)| 0.0019 (0.0008) |
> |0.25 | 0.0849 (0.0016) | 0.2669 (0.0022) | 0.1389 (0.0021) | 0.1405 (0.0028)| - |
> |0.5 | 0.1204 (0.0036) | 0.3017 (0.0043) | 0.1619 (0.0027) | 0.1660 (0.0025)| - |
> |0.75 | 0.1497 (0.0046) | 0.3433 (0.0067) | 0.1925 (0.0061) | 0.1821 (0.0336)| - |
> |1 | 0.1871 (0.0071) | 0.3884 (0.0094) | 0.2265 (0.0035) | 0.2381 (0.0049)| 0.0272 (0.0014)|
> |1.25 | 0.2280 (0.0053) | 0.4250 (0.0062) | 0.2824 (0.0043) | 0.2872 (0.0065)| - |
> |1.5 | 0.2676 (0.0059) | 0.4634 (0.0085) | 0.3360 (0.0042) | 0.3304 (0.0104)| - |
>
> We rename SS Ablated from the manuscript to AttrPrompt to emphasize the similarity of the ablation to AttrPrompt and the role it serves as a baseline approach. We add two additional ablation requested by other reviewers:
> - SimpleStrat w/o 20-Questions ablates the 20 questions approach keeping everything else the same
> - SimpleStrat Single Prompt ablates the need for multiple stages. We append the prompts for the three stages into a single stage and give the model a 1000 token prompt and a budget of 1000 tokens to follow the instructions. We chose to only sample temperature 0 and temperature 1 as the results show substantially worse performance than directly prompting gpt-4o.
>
> The ablations further support the claim that 20-Questions formulation leads to far better partitions and is crucial to the observed improvements. The single prompt ablation supports the importance of having multiple stages. In particular, the single prompt is missing probabilistic prompting severely limiting the diversity. The standard deviation gives us confidence that the gains of full SimpleStrat are not the result of sampling variance.
>
> ### Baseline to increase sample diversity:
> We disagree that no baselines aimed at sample diversity were included. We argue temperature scaling is the most common and widely accepted baseline to `increasing sample diversity`. We show just applying SimpleStrat to temperature zero gives as much benefits to diversity as increasing the temperature to 1.2 in CoverageQA and 1 in WritingPrompts. We not only show absolute improvement over temperature scaling but orthogonal gains that show even at temperature 1.5 we can improve the recall on CoverageQA from 0.35 to 0.45. We further compare to AttrPrompt (Yu et al 2023) labeled SS Ablation which we believe to be the most comparable baseline also introducing variance by sampling from a family of prompts.
>
> Zhu et al. [1] and Garces Arias et al.,[2] will naturally shine in settings with longer generations as it selectively chooses where to spend “temperature” based on model confidence. In the short answer setting of CoverageQA, these approaches are similar to just temperature scaling as there are few (5-10) tokens in the generation. Moreover, as illustrated in Figure 1 and Figure 7, the instruction-tuned models tend to be poorly calibrated in terms of next-token probability. In the context of the WritingPrompts experiment, the goal is to highlight the degradation of quality at higher temperatures. Thus, SimpleStrat accesses a different kind of variance than accessible using model confidence.
>
> We discuss LMX[3] in line 127 of the related works. This approach relies on carefully choosing a few-shot examples from a seed set of diverse examples to extract variance. Our approach does not rely on external guidance by providing few-shot examples. A natural extension of SimpleStrat to consider is “sampling without replacement” where past samples can be leveraged to generate new unique samples. In the “without replacement” setting, SimpleStrat can be extended in 3 key ways:
> 1. AutoStratification can be used to introduce partitions that distinguish already seen solutions from not yet seen solutions,
> 2. AutoStratification can repartition the remaining unobserved solution space,
> 3. Probabilistic Prompting can upweighting partitions that have produced less than expected samples.
> In this setting LMX and SimpleStrat can be viewed as complementary mutators for discovering new solutions. In summary, LMX cannot be directly compared to SimpleStrat as it relies on a strong seed bank of solutions.

---

> > ### Author Response · Authors · 2025-08-04
> >
> > We hope our rebuttal has clarified the questions raised in the review. If you have any further questions or concerns, we’re happy to follow up.

---

> > ### Comment · Reviewer_VKJt · 2025-08-05
> >
> > Thank you for your response. I believe clarifying the points in the rebuttal in the revised manuscript should be helpful for enhancing its quality.
> >
> > **Regarding stronger baselines**
> >
> > 1. Temperature scaling is a trivial baseline compared to the methods that are specifically designed for improving sampling diversity proposed in the related work I mentioned. In my review, I was referring to comparisons with those baselines.
> >
> > 2. It is helpful to know that the SS ablation corresponds to AttrPrompt (Yu et al., 2023). This should be made clearer in the revised manuscript. The original version did not mention this, and Yu et al. is only briefly discussed.
> >
> > 3. Overall, I believe that more thorough empirical comparisons with related methods for improving sampling diversity should be considered to better demonstrate the significance of the proposed approach. Given this, I will maintain my original assessment.

---

> > > ### Author Response · Authors · 2025-08-05
> > >
> > > Thanks for reading the rebuttal and providing feedback on our work.
> > >
> > > **Clarification on Baselines.** We will add the clarifying points to the revised manuscript as we agree it will enhance the clarity of the paper. Specifically, 1) we will clarify in the manuscript that the SS Ablation corresponds to the baseline AttrPrompt (Yu et al., 2023) and more carefully discuss how AttrPrompt shares a probabilistic prompting approach as in SimpleStrat, and 2) we will also add the additional ablations on the "20 questions" formulation show the quality of the constraints is crucial to the improvements observed. Along with temperature scaling, these two baseline show SimpleStrat outperforms the common approaches to improving diversity. Further, when combined with temperature-based approaches leads to more diversity across all temperatures.
> > >
> > > We appreciate questions you raised as the clarifications will improve the manuscript. We further agree that diverse generation is a important problem with practical value, and that our work shows that SimpleStrat outperforms baselines in a variety of settings.

---

### Official Review · Reviewer_ruyA · 2025-07-21

**Clarity:** 3
**Significance:** 3
**Originality:** 3
**Rating:** 5
**Confidence:** 4

**Summary:**

# Problem
Generating meaningfully diverse outputs from large language models (LLMs) remains a challenge, especially when diversity must be semantically significant. Existing techniques like temperature scaling or context-based prompting have shown limited success in ensuring genuine response variety.

# Contributions
This paper makes three main contributions:

- SimpleStrat - a training-free pipeline for controllable, interpretable diversification of LLM outputs that operates entirely at inference time.
- CoverageQA - a dataset of underspecified questions designed to evaluate response diversity.
- New diversity metrics - tailored for different evaluation settings (with or without access to model logits), aimed at capturing distinct types of diversity.

# Method
The SimpleStrat approach consists of three stages:

- Automatic stratification - automatically identifies meaningful diversity dimensions within the solution space.
- Heuristic estimation - estimates the relative proportions of identified strata.
- Probabilistic prompting - samples prompts based on heuristic-driven distributions, balancing randomness with controlled diversity.

SimpleStrat enhances diversity without the need for model fine-tuning and avoids the quality degradation often seen with naive sampling strategies like high-temperature generation.

# Experimental Setup
Recognizing that existing datasets and metrics fall short for evaluating diversity, the authors created CoverageQA in two variants:

- CoverageQA-Curated (manually curated set of underspecified questions)
- CoverageQA-Wikipedia (derived from Wikidata)

They also introduce three diversity metrics:

- Distributional Diversity — for models with access to token probabilities (KL-based).
- Coverage Diversity — for black-box models, based on sampling over CoverageQA (Recall-based).
- Open-Ended Diversity — for tasks with inherently unenumerable answer spaces (Embedding distance-based)

# Results

- SimpleStrat outperforms GPT-4o and Claude 3.5 Sonnet on coverage diversity.
- SimpleStrat achieves higher distributional diversity than LLaMA 3 (direct comparisons to GPT-4o and Claude were infeasible for this metric due to sampling costs).
- SimpleStrat also surpasses GPT-4o in terms of open-ended diversity.

**Questions:**

# Major Questions and Comments

I strongly encourage the authors to address the points below. I will consider increasing the quality score and overall rating if the following issues are adequately addressed:
1. Statistical significance of the results: Regarding the experiments in Figure 4, only mean results are reported, with no indication of variance. A proper evaluation can be one of the following:
     (a) Resampling the set of 100 inputs multiple times, computing recall for each, and reporting standard error (or standard deviation) to demonstrate the robustness of the results with respect to sampling variability.
     (b) If resampling is computationally/monetary expensive for 100 inputs (as in (a)), this limitation should be explicitly stated. In that case, you could reduce the number of input questions (e.g., to 20–25) and increase the number of sampling runs (e.g., 4–5) to better estimate variability.
     (c) Additionally, bootstrapping (sampling with replacement) from the original inputs could be used to generate pseudo-samples and estimate variability. The computational/financial limitation should be explicitly stated as well in this case.

I encourage the authors to explore which of these options are feasible within their available resources.
2. Figure 5: The same argument applies to the results in Figure 5. Assessing robustness is not possible when only mean values are reported. Please report standard error or standard deviation for these results as well.
3. Table 1:The same applies to the distributional diversity results in Table 1. Please report standard error or standard deviation to support your findings.
4. Use of open-source models: Are there any open-source models (at least one more) that you could include in the distributional diversity experiments to strengthen your evaluation?
5. Consistency with NeurIPS checklist (Q7): In light of points (1)–(4), I find your response to Question 7 of the NeurIPS checklist regarding statistical significance somewhat inconsistent with the reported results. Please verify consistency and adjust text/answer accordingly.
6. Limitations of the work:  Unless I have overlooked it, I suggest explicitly stating the limitations of your work in the main text. This could be done in a dedicated paragraph or section, possibly incorporating your response to Question 2 of the NeurIPS checklist.
7. Code availability (Checklist Q4, Q5, Q13):  Could the authors clarify whether their responses to Checklist Questions 4, 5, and 13 refer to system prompts provided in the Appendix or the entire experimental pipeline (I have not found reference to the later one)?


# Minor Questions and Comments

I will consider improving the quality score and overall rating if the following questions and comments are addressed. However, I strongly encourage the authors to prioritize the major comments first.

1. Line 7: “Partition of space...” - Please clarify which space is being referenced (e.g., solution space).
2. Lines 33 - 34: Please consider reformulating this sentence. While it is understood that increasing temperature leads to a less skewed distribution, the current phrasing is difficult to comprehend.
3. Lines 41 - 42: Please consider citing relevant research from cognitive or social sciences that formally demonstrates the presence of the "blue-seven" phenomenon.
4. Line 47: The phrase “aligned with true distribution of answers” appears overly strong, given that no experiments in the paper directly validate this claim. Moreover, for some experiments (such as open-ended questions), it may be impossible to compute this distribution.
5. Lines 68 - 76: I suggest introducing all three diversity types here. For example, readers may not immediately associate the "recall" measure with "coverage diversity," as introduced later. The same applies to "text-embedding distance" and "open-ended diversity."
6. Line 71: The phrase “recall being the most natural way” may come across as an overstatement. Consider avoiding superlative forms, though this is optional.
7. Line 79: The value 0.36 is difficult to interpret, as KL divergence is only guaranteed to be non-negative and has no defined upper bound. Please consider providing additional reference values or context to help readers assess the significance of this figure. The same issue arises on line 96.
8. Line 92: Consider adding “of output” or “of response” after the word “diversity” for clarity.
9. Line 141: The sentence is somewhat difficult to read. Please clarify - the validity of what? Also, why are augmentations described as “more aggressive”?
10. Line 198: Should this read “all **solutions** in the solution space”?
11. Line 211: A closing parenthesis “)” appears to be missing.
12. Lines 230 - 235: I recommend providing a clear example of how the method generates outputs. While I appreciate the authors’ efforts in the main text and Appendix A, the description remains somewhat abstract. A concrete example would be helpful (possible in the Appendix, but step-by-step).
13. Figure 3 caption: I assume the left side shows GPT-4o results and the right side shows SimpleStrat. Please ensure the caption consistently reflects the figure’s layout.
14. Figure 4 legend: Please clarify what is meant by “baseline.” Is it GPT-4o?
15. Figure 4 legend: Please use “Claude 3.5 Sonnet” instead of “Anthropic” and “GPT-4o” instead of “baseline” for consistency.
16. Figure 4 caption: Could you please elaborate on what exactly is meant by “SS Ablation Recall”?
17. Figure 4 caption: I suggest adding a brief explanation for why the evaluation of Claude 3.5 Sonnet is limited to temperatures up to 1.0. The same clarification would be helpful for Figure 8 and Figure 9a.
18. Figure 5: While this is a matter of presentation preference, since both plots are placed close together, I suggest using the same colors for curves representing the same model to avoid confusion. Currently, blue and orange denote different models in each plot.
19. Line 286 - 287: The phrase “These simple checks are do not aspire” contains a grammatical error. Please revise.
20. Line 297: Please correct “Furthre” to “Further.”
21. Line 326: In Figure 6, the x-axis represents SimpleStrat and the y-axis represents the baseline. Please ensure that the text and figure labeling are consistent. 22. Plans for CoverQA release: Finally, are there any plans to release CoverQA? For example, as CoverQA-Curated is intended to remain unchanged, could it be published "as is"? Could CoverQA-Wiki be snapshotted and published (while being continuously updated online)? This is important, as it is one of the claimed contributions.

**Ethical Concerns:**

["NO or VERY MINOR ethics concerns only"]

**Final Justification:**

The authors have adequately addressed my concerns; consequently, I have increased the quality score and overall ranking to reflect the improvements made during the rebuttal phase.

**Limitations:**

The authors provided clarification regarding the limitations of their work in the NeurIPS checklist; however, there is no dedicated section or paragraph explicitly discussing these limitations. In my comments, I recommended placing greater emphasis on this point within the "Questions" section.

**Paper Formatting Concerns:**

I did not identify any formatting issues.

**Quality:**

3

**Strengths And Weaknesses:**

# Strengths

1. Originality: I found the core idea to be simple and straightforward, yet original. The proposed method enables increasing the diversity of LLM outputs in an interpretable and controllable manner (at the inference stage).

2. Significance: Given the widespread use of LLMs and the large number of downstream applications where users require valid, faithful, yet semantically diverse generated content, I consider the significance of this work to be high.

3. Clarity: The manuscript is well-structured and well-written. The problem statement and research gap are clearly articulated, and the subsequent ideas are logically developed and easy to follow (apart from a few points which I have listed in the "Questions" section.)


# Weaknesses

1. Quality: While I found the paper to be of reasonable quality overall, I have significant concerns regarding the lack of statistical significance in the results and the limited scope of the evaluation. Additionally, there are several minor issues scattered throughout the manuscript that I encourage the authors to address (please see the "Questions" section).

---

> ### Author Rebuttal · Authors · 2025-07-29
>
> We appreciate the detailed review and the constructive feedback! We hope the following clarifications and experiments address your questions.
>
> ## Major Questions
> We designate answer to major questions with **(Q#)**
>
> ### On code availability and dataset availability **(Q7)**:
> As part of the submission, we included a zip supplemental file including the experiment pipeline as well as plotting scripts. We include the prompts in the appendix of the manuscript for easier access and legibility. Additionally, we provided the CoverageQA datasets and raw inference outputs from the language models in this zip file. In particular, we want to highlight that the CoverageQA evaluation pipeline requires careful handling of redundant ways of expressing the same answer – for instance countries (e.g. "United Arab Emirates","UAE", and "Emirates") as well as for names (e.g. "J. Robert Oppenheimer", "Robert Oppenheimer", "Oppenheimer", and "J. R. Oppenheimer"). Our inclusion of the experiment pipeline in the supplemental is important for replicability given the need for handling these duplicates.
>
> In summary, CoverageQA has already been made available in the submission. As noted by the reviewer, CoverageQA is procedurally generated and thus will be periodically updated with future versions based on changes to WikiData.
>
> ## Experiment robustness (**Q1-Q3**),
> As recommended, we reran the results in figure 4 and 5 to obtain statistical confidence. In the original experiment, we focused our compute/cost budget on running a wider variety of temperature and large number of resampling to provide confidence that the results are not due to randomness of samples. As requested, below are the results of replicating the figure 4 CoverageQA experiment on a smaller subset of temperatures (to reduce cost) repeated 5 times. They confirm the results are robust to sampling:
>
> ### CoverageQA Results with std and additional Ablations -- Original Figure 4 **(Q1)**
>
> | Temperature | Baseline GPT-4o (std) | SimpleStrat with GPT-4o (std) | AttrPrompt with GPT-4o (std) | SimpleStrat w/o 20-questions|
> |-------------|-----------------------|-------------------------------|---------------------------------------|--|
> |0 | 0.0646 (0.0011) | 0.2423 (0.0050) | 0.1244 (0.0006) | 0.1215 (0.0021)|
> |0.25 | 0.0849 (0.0016) | 0.2669 (0.0022) | 0.1389 (0.0021) | 0.1405 (0.0028)|
> |0.5 | 0.1204 (0.0036) | 0.3017 (0.0043) | 0.1619 (0.0027) | 0.1660 (0.0025)|
> |0.75 | 0.1497 (0.0046) | 0.3433 (0.0067) | 0.1925 (0.0061) | 0.1821 (0.0336)|
> |1 | 0.1871 (0.0071) | 0.3884 (0.0094) | 0.2265 (0.0035) | 0.2381 (0.0049)|
> |1.25 | 0.2280 (0.0053) | 0.4250 (0.0062) | 0.2824 (0.0043) | 0.2872 (0.0065)|
> |1.5 | 0.2676 (0.0059) | 0.4634 (0.0085) | 0.3360 (0.0042) | 0.3304 (0.0104)|
>
> We rename SS Ablated from the manuscript to AttrPrompt to emphasize the similarity of the ablation to AttrPrompt and the role it serves as a baseline approach. We add two additional ablation requested by other reviewers:
> - SimpleStrat w/o 20-Questions ablates the 20 questions approach keeping everything else the same
> - SimpleStrat Single Prompt ablates the need for multiple stages
>
> The ablations further support the claim that 20-Questions formulation leads to far better partitions and is crucial to the observed improvements. The single prompt ablation supports the importance having multiple stages. In particular, the single prompt is missing probabilistic prompting severely limiting the diversity. The standard deviation gives us confidence that the gains of full SimpleStrat are not the result of sampling variance.
>
> ### WritingPrompts Results with std -- Original Figure 5 **(Q2)**
>
> As recommended, we report the standard deviation of the metrics over the 100 samples for each temperature setting. The generations require producing full outlines of stories based on creative writing prompts from reddit, WritingPrompt. This reported variance ensures against sampling variance over the set of prompts taken from the full writing prompts dataset.
>
> **Baseline GPT-4o**
> | Temperature | Diversity – cosine distance (std) | Quality – Format Adherence (std)
> |-------------|-----------------------|------------|
>  | 0.0 | 0.0007 (0.0043) |  1.0000 (0.0000) |
>  | 0.25 | 0.3897 (0.1320) |  0.9700 (0.1706) |
>  | 0.5 | 0.4537 (0.1281) |  1.0000 (0.0000) |
>  | 0.75 | 0.5171 (0.1050) |  0.9500 (0.2179) |
>  | 1.0 | 0.5511 (0.1057) |  0.9500 (0.2179) |
>  | 1.25 | 0.7097 (0.1815) |  0.3400 (0.4737) |
>
>
> **SimpleStrat with GPT-4o**
> | Temperature | Diversity – cosine distance (std) | Quality – Format Adherence (std)
> |-------------|-----------------------|------------|
>  | 0.0 | 0.5444 (0.1664) |  0.9900 (0.0995) |
>  | 0.25 | 0.5627 (0.1723) |  1.0000 (0.0000) |
>  | 0.5 | 0.5979 (0.1434) |  0.9900 (0.0995) |
>  | 0.75 | 0.6176 (0.1211) |  0.9700 (0.1706) |
>  | 1.0 | 0.6420 (0.1142) |  0.9300 (0.2551) |
>  | 1.25 | 0.7535 (0.1465) |  0.3100 (0.4625) |
>
> We emphasize that SimpleStrat is able to randomize in the prompt space even for temperature zero. As seen the diversity as measured by cosine distance in the embedding space is significantly larger than without SimpleStrat and is comparable to much higher temperatures. It’s interesting to note that even with temperature = 0 on longer generation tasks the generated outlines have small deviations.
>
> **(Q3)**: For results on distributional diversity, the constraints are generated with temperature 0 and the next token probabilities are computed deterministically from the model weights. As such, there's no natural notion of sample variance. **(Q5)** We will update the checklist in the appendix to clarify which experiments we provide standard deviations for.
>
> ### On open source models **(Q4)**:
> At the time of selection, we chose Llama 3.1 as it was the state-of-the-art open-weight model. We believe the learnings are general as the observation of existing mode collapse and improvements by SimpleStrat are consistent with analysis on proprietary models. We will include Qwen-3 as well in the final revision as it has grown in popularity and capability. We do not anticipate significant differences in results as new open-source models get released since the proprietary models also show the same limitation. Particularly, we anticipate mode-collapse in instruction-tuned models may become even more severe in reasoning models where models have been trained on objective single-answer math problems. We leave analysis of reasoning focused models as future work.
>
> ### On limitations **(Q6)**,
> In section 3.4, We acknowledge the risk that SimpleStrat identifies “unwanted bias or unwanted factors” when proposing strata and proportions. Specifically, we mention that even in the case that the LLM correctly follows the 20 questions-inspired instruction to divide the space of unique solutions into equal proportions, this may have unintended consequences. For instance, there are more female baby names than male baby names as reported by the census. In that case, SimpleStrat would exhibit a preference for female baby names. We will include a dedicated section to emphasize the potential limitation of relying on model produced stratification in the main text.
>
> To further elaborate, although SimpleStrat shows improvement empirically, it is sensitive to the model selecting good meaningful axes in auto-stratification and correct joint probabilities in heuristic estimation. As work on LLMs for forecasting improves, we expect LLMs to produce better estimates especially when given access to external data and data analysis tools. For our prototype, we restricted ourselves to studying the model’s intrinsic capabilities and tendencies. Further, the model may have biases concerning race and gender that may be reflected also in the auto-stratification and heuristic estimation. As such, it is recommended the probabilistic prompt distribution of SimpleStrat is carefully inspected for critical applications. Finally, CoverageQA is a dataset of short responses. Although this makes evaluation more practical, we believe SimpleStrat will be especially impactful in settings that require low temperature, especially multi-step reasoning as identified by Zhang et al. (2024).
>
> ----------------------------------------------
> ## Other questions:
> ### SimpleStrat Ablation (SS Ablation):
> To clarify, the abbreviation SS Ablation refers to the ablation described in section 5.1 under coverage diversity. The ablation asks the language model directly to propose criterion instead of the 20-questions approach in AutoStratification and the ablation applies uniform sampling akin to AttrPrompt (Yu et al 2023) instead of using heuristic estimation. We will make this explicit in the caption as recommended.
>
> ### Abbreviated Legend in Figure 4:
> Indeed Baseline indicates using GPT-4o without SimpleStrat, Anthropic stands for Claude-3.5 sonnet, SS stands for SimpleStrat, SS Ablation stand for the Ablation described above where we replace the 20-questions approach to AutoStratification with a direct prompt asking for partitions. We will make the recommended changes to the manuscript make sure the legend is clear and precise.
>
> ### Anthropic Model Temperatures:
> Footnote 3 for figure 4 explains that Claude models are only allowed temperatures between 0 and 1; the footnote unfortunately rendered on page 9. This is possibly because alignment of models is more difficult to guarantee for high temperatures.
>
> ### Blue-Seven Phenomena:
> We will include a citation for blue-seven and raise a caveat that the effects may vary across cultures: Simon, William E. "Number and color responses of some college students: Preliminary evidence for a “blue seven phenomenon”." Perceptual and motor skills 33.2 (1971): 373-374.

---

> ### Comment · Reviewer_ruyA · 2025-08-04
>
> Dear Authors,
>
> Thank you for addressing my comments. Please ensure that the promised revisions are incorporated into the final version of the paper - please pay particular attention to reporting statistical significance and explicitly discuss limitations in the main text. Conditional on the faithful implementation of these changes, I have raised the quality score from 2 to 3 and the overall ranking from 4 to 5.

---

> ### Author Response · Authors · 2025-08-04
>
> Thanks for revisiting the review! We will implement the changes discussed in the rebuttal we described and your detailed comments on smaller changes to make the manuscript clearer.

---

### Note · Authors · 2025-08-14

## Reviewer Response – SimpleStrat
First, we thank the reviewers for detailed suggestions and enthusiastic reception. We appreciate that reviewers highlighted:
1. **Originality.** SimpleStrat takes an intuitive yet original approach—treating semantic diversity like a game of **20 questions** to automatically generate partitions of the solution space.
2. **Effectiveness.** Improvement to semantic diversity on baselines across multiple settings — namely, multi-answer QA (CoverageQA) and creative writing (WritingPrompts) spanning proprietary (Claude-3.5, GPT-4o) and open-weight (Llama-3.1) models.
3. **Significance.** Addresses an under-explored yet impactful direction in LLM research: systematically improving **diversity and coverage**.

## Restated Contributions

- **SimpleStrat**: A training-free method for improving diversity via auto-stratification, heuristic estimation, and probabilistic prompting—controllably adding randomness at the prompt level.
- **CoverageQA Dataset**: A benchmark of **multi-answer questions**, enabling recall-based evaluation as a proxy for semantic diversity.
- **Three Diversity Metrics**:
  1. *Distributional Diversity*: KL divergence from the expected answer distribution (open-weight models)
  2. *Coverage Diversity*: Recall over expected answers (proprietary models)
  3. *Open-ended Diversity*: Embedding cosine distance

## Additional Ablations – *CoverageQA* Average Recall
We clarified that the “SS Ablation” setting is **equivalent to AttrPrompt(Yu et al 2023)**, the most relevant baseline from prior work uniform sampling w/o 20-Q.

We also provided two ablations to isolate the components of SimpleStrat:
- **No 20‑Q Abl.**: Significantly worse performance, highlighting the central role of 20-questions for semantic partition quality.
- **Single‑Prompt Abl.**: Using the 20-Q prompt without probabilistic prompting performs worse than simply raising generation temperature.

| Temp. | Just GPT-4o (std) | SimpleStrat | AttrPrompt | No 20-Q Abl. | Single-Prompt Abl.
|-|-|-|-|-|-|
|0 | 0.0646 (0.0011) | **0.2423 (0.0050)** | 0.1244 (0.0006) | 0.1215 (0.0021)| 0.0019 (0.0008)
|1 | 0.1871 (0.0071) | **0.3884 (0.0094)** | 0.2265 (0.0035) | 0.2381 (0.0049)| 0.0272 (0.0014)

**Takeaways:**
- *20‑questions* is essential—removing it degrades recall close to baseline levels.
- SimpleStrat meaningfully outperforms in both low‑ and high‑temp regimes.
- Probabilistic prompting is critical; without it, performance collapses.

---

### Decision · Program_Chairs · 2025-09-17

**Decision:**

Accept (poster)

**Comment:**

This work introduces the method SimpleStrat which is designed to increase diversity from LLM output. The approach uses stratified sampling from auto-generated strata produced by the LLM. The method is evaluated on CoverageQA, a benchmark that is introduced by the authors to measure response diversity. The approach performs well on the benchmarks.

Strengths:
* The paper addresses an important underexplored research area, sampling diverse responses from LLMs.
* The method is easy to understand explained well.
* The work introduces a new benchmark CoverageQA to assess response diversity.
* Several metrics are measured including KL divergence, recall and embedding distance.

Weaknesses:
* The approach relies on the ability of the LLM to generate good strata through prompting. This may not be effective for certain applications such as generating answers to math questions which may have multiple valid answers.
* There is a significant amount of computational overhead involved in generating strata and heuristic estimation. The authors state this can be cached but that is still going to be costly if only generating a few samples for a given user request and requires the user to explicitly use this method for generation.
* The evaluation relies on automatic constraints from Wikidata so doesn't represent a comprehensive evaluation of diversity. Measurements by humans would help.

Due to the novelty of approach, the benchmark contribution and the likely significant impact on work that explores sampling diversity, I recommend that this work be accepted.

During the discussion, reviewers asked for more careful ablations, including whether the 3 stages are needed and whether the 20 questions approaching to stratification were needed. The authors successfully addressed these questions by adding additional ablations. This was helpful for the final decision.